# Unfolding and identification of membrane proteins in situ

**Nicola Galvanetto**[1]*[†‡], **Zhongjie Ye**[1†], **Arin Marchesi**[2,3], **Simone Mortal**[1], **Sourav Maity**[1,4], **Alessandro Laio**[1,5], **Vincent Torre**[1,6,7]

[1]International School for Advanced Studies, Trieste, Italy; [2]Nano Life Science Institute, Kanazawa Medical University, Kanazawa, Japan; [3]Department of Experimental and Clinical Medicine, Università Politecnica delle Marche, Ancona, Italy; [4]Moleculaire Biofysica, Zernike Instituut, Rijksuniversiteit Groningen, University of Groningen, Groningen, Netherlands; [5]The Abdus Salam International Centre for Theoretical Physics, Trieste, Italy; [6]Institute of Materials (IOM-CNR), Area Science Park, Trieste, Italy; [7]BioValley Systems & Solutions, Trieste, Italy

**\*For correspondence:**
nicola.galvanetto@sissa.it

[†]These authors contributed equally to this work

**Present address:** [‡]University of Zurich, Zurich, Switzerland

**Competing interest:** The authors declare that no competing interests exist.

## Abstract

Single-molecule force spectroscopy (SMFS) uses the cantilever tip of an atomic force microscope (AFM) to apply a force able to unfold a single protein. The obtained force-distance curve encodes the unfolding pathway, and from its analysis it is possible to characterize the folded domains. SMFS has been mostly used to study the unfolding of purified proteins, in solution or reconstituted in a lipid bilayer. Here, we describe a pipeline for analyzing membrane proteins based on SMFS, which involves the isolation of the plasma membrane of single cells and the harvesting of force-distance curves directly from it. We characterized and identified the embedded membrane proteins combining, within a Bayesian framework, the information of the shape of the obtained curves, with the information from mass spectrometry and proteomic databases. The pipeline was tested with purified/reconstituted proteins and applied to five cell types where we classified the unfolding of their most abundant membrane proteins. We validated our pipeline by overexpressing four constructs, and this allowed us to gather structural insights of the identified proteins, revealing variable elements in the loop regions. Our results set the basis for the investigation of the unfolding of membrane proteins in situ, and for performing proteomics from a membrane fragment.

## Editor's evaluation

This paper presents a method to identify membrane proteins in native cell membranes based on a combination of single molecule AFM and an unsupervised clustering procedure to identify clusters of single-protein curves. This original approach represents a definitive step forward for AFM technology and methodology, which can generally only be used to characterize purified biomolecules of known identity.

## Introduction

Mapping and recovering the structure of membrane proteins is a challenging aim. Arguably, the most successful tool for this purpose is cryo electron microscopy (cryo-EM) – but it requires frozen samples, and precise single particle measurements can be achieved only with purification. Cryo-EM and now AlphaFold *Jumper et al., 2021* have changed structural biology since the molecular structure of possibly all proteins can be determined with reasonable effort. However, these tools determine only

the most probable structure and – at the moment – are not able to identify the various configurations visited by the proteins or their mechanical properties in physiological conditions.

In fact, much of what we know about the mechanics and the structure at room temperature (RT) of cell membranes and membrane proteins comes from atomic force microscopy (AFM) (*Al-Rekabi and Contera, 2018*; *Casuso et al., 2012*; *García-Sáez et al., 2007*; *Zuttion et al., 2018*) which can operate in liquid environments with nanometric resolution.

AFM-based single-molecule force spectroscopy (SMFS) uses the tip of an AFM to apply a stretching force to unfold a single protein. SMFS provides a 1D force-distance (F-D) curve which encodes the unfolding pathway so that from the analysis of the sequence of force peaks it is possible to identify the folded domains and their variability (*Engel and Gaub, 2008*). It has been recently confirmed that from the 1D amino acid sequence of a protein it is possible to accurately determine its 3D structure (*Jumper et al., 2021*), therefore it is tempting to continue exploring which information can be recovered from totally different 1D spectra – the F-D curves provided by SMFS.

Hitherto SMFS has been mostly used to study the unfolding mechanics of purified proteins, in solution or reconstituted in lipid bilayers. Although the information that is possible to obtain at room temperature (RT) is of great interest (e.g. it allows studying mechanical stability [*Sumbul et al., 2018a*; *Thoma et al., 2015*] or structural heterogeneity [*Hinczewski et al., 2016*]), unfolding experiments have been performed only in less than 20 different membrane proteins in the last 20 years, most likely because of the difficulties involved in purification and reconstitution. Moreover, studying membrane proteins in their native membrane instead of purifying them would be preferable because their folding state highly depends on the physical and chemical properties of the cell membrane and on their molecular partners that might cooperatively function nearby (e.g. in case of oligomers; *Maity et al., 2015*; *Thoma et al., 2018*).

In this manuscript we aim to bridge the gap between all these recent breakthroughs by attempting to identify and recover structural information of membrane proteins embedded in biological samples, namely in their native environment. In this way we would like to obtain information on mechanical properties and on the possible structural heterogeneity at RT of a wide range of proteins, overcoming the limiting factor of purification that hindered the wide application SMFS on membrane proteins.

We describe a complete pipeline, including the experimental methods and the data analysis, which represent a first step forward in this direction. The pipeline allows to identify membrane proteins obtained from SMFS on single cells. First, we developed a technique to extract a piece of the membrane suitable for SMFS so to obtain millions of F-D curves from native biological membranes. Second, we used an unsupervised clustering procedure to detect sets of similar unfolding curves. Finally, we implemented a Bayesian meta-analysis using information from mass spectrometry and protein structure databases that allows to identify a limited list of candidates of the unfolded proteins, which can then be confirmed with specifically engineered constructs. We first characterized the unfolding of purified membrane proteins reconstituted in vitro in lipid bilayers. Then, we focused on the native cell membranes of five cell types (primary cells and cell lines).

Our protocol allowed us to obtain the combined protein profile of the isolated membrane fragments, and to gather specific structural information on variable segments of the identified proteins. We expanded the number of known unfolding spectra by more than 40, and we provided the molecular identification of four mammalian membrane proteins. Unexpectedly, the distribution of the membrane protein population found with mass spectrometry on thousands of cells can also be recovered with our F-D curves obtained from 3 to 10 cells, suggesting that membrane proteomic may be possible at the single-cell level.

Even if the proposed protocol cannot compete with the identification accuracy of mass spectroscopy performed on thousands of cells, or with the data quality of a SMFS experiment performed on a single purified protein, it accomplishes a different task – it allows gathering biologically meaningful information (i.e. structural properties at RT and in the native cell membrane, protein profiling) at the single-cell level that cannot be captured otherwise.

## Results

### Unfolding proteins from isolated cell membranes

The first ingredient of our pipeline is an unroofing method (*Galvanetto, 2018a*) to isolate the apical part of single-cell membranes (see *Figure 1—figure supplement 1*) containing membrane proteins with negligible contamination of cytoplasmic proteins (*Figure 1—figure supplements 2–4*). We sandwiched a single cell between two glass plates, the culture coverslip and another plate mounted on the AFM itself (see *Figure 1B–C*, triangular coverslip). The triangular coverslip is coated with poly-lysine which favors membrane adhesion. A rapid separation of the plates permits the isolation of the apical membrane of the cell (see *Figure 1D–E*). The method is reliable (*n*=105, ~80% success rate) with cell types grown on coverslips. With non-adherent cells, like freshly isolated rods, membranes were isolated with a lateral flux of medium (*Clarke et al., 1975*) (see Methods). During the unroofing process, we verified the absence of any adsorption of cytoplasmic material like tubulin, actin, mito-chondria, or free cytoplasmic proteins from the unroofed membrane patch where only the membrane proteins that are hold by the lipids are present (see *Figure 1—figure supplements 3–5*).

We verified with the AFM (*Figure 1F*) that the isolated membrane patches have a height of 5–8 nm with roughness in the order of 1 nm. Then, we performed conventional SMFS (*Oesterhelt et al., 2000*) collecting more than 2 millions of F-D traces from all the samples (hippocampal neurons, DRG neurons, neuroblastoma, rods, and rod discs, see below). Among the obtained curves, ~95% shows no binding (*Figure 1G*), ~3% shows plateau ascribable to membrane tethers (*Figure 1H*), while the remaining ~2% displays the sawtooth-like shape that characterizes the unfolding of proteins (*Oesterhelt et al., 2000*; *Figure 1I*) with each tooth fitting the worm-like chain (WLC) model with a persistence length of ~0.4 nm (*Figure 1—figure supplement 6*; *Li et al., 2002*). We point out that for membrane proteins, the term *unfolding* refers broadly to the tertiary structure, because the specific timing and dynamics of the pullout of a transmembrane segment and the unfolding of its secondary structure is not resolved yet. The AFM tips becomes attached to membrane proteins mainly by hydrophobic and hydrophilic interactions (*Müller and Engel, 2007*). For purely kinetic reasons it will be more likely the AFM will get attached to the C-terminus or the N-terminus, since those typically exert less resistance to traction than a loop. In other words, if the tip gets attached to a loop, the trace will be very short, and not detected by our analysis. This mechanism is non-specific and agnostic of the specific nature of the protein. We decided not to functionalize in any manner the AFM tip in order to avoid to introduce biases. These ~50,000 traces are dramatically hetero-geneous, differing in total length, number of peaks, maximum force, etc. This is not surprising, since cell membranes contain a large number of different proteins, each hosted in different local environment. In the following we describe a procedure which allows, to some extent, disentangling this bundle.

In SMFS it is assumed that the binding between the cantilever and the protein occurs either at the C- or at the N-terminus, and that the protein is fully unfolded by the tip. However, some traces suggest that also other events take place: (i) the simultaneous attachment of two or more proteins to the tip (*Walder et al., 2017*), (ii) the incomplete unfolding of the attached protein (*Tanuj Sapra et al., 2006*), (iii) the binding of the AFM tip to a loop of the protein instead of to a terminus end (*Figure 1—figure supplement 7C-F*). (i) *Attachment of multiple proteins* (*Figure 1—figure supplement 7D*): the resulting F-D curves will not have a recurrent pattern; if two proteins form a complex, the resulting spectrum is the sum of the two individual spectra, which causes deviations of the measured persistence length (*Figure 1—figure supplement 8*). The simultaneous unfolding of multiple proteins is also characterized by force changes and varying persistence length (*Figure 1—figure supplement 7D,G* and *Figure 1—figure supplement 8*). (ii) *Incomplete unfolding of the protein* (*Figure 1—figure supplement 7E*): if the tip prematurely detaches from the terminus, the F-D curve has similar but shorter pattern compared to a complete unfolding (*Figure 1—figure supplement 7C*). The fraction of curves that prematurely detaches has been reported to be ~23% of the fully unfolded protein (*Tanuj Sapra et al., 2006*).

(iii) *Binding of the AFM tip to a loop* (*Figure 1—figure supplement 7F*): this case is equivalent to the attachment of multiple proteins. However, if the attachment of the cantilever tip to a loop occurs with some consistency, we will obtain a recurrent pattern with the features described in case (i) (devi-ation of persistence length during intersection, two major levels of unfolding force, see *Figure 1—figure supplement 8*).

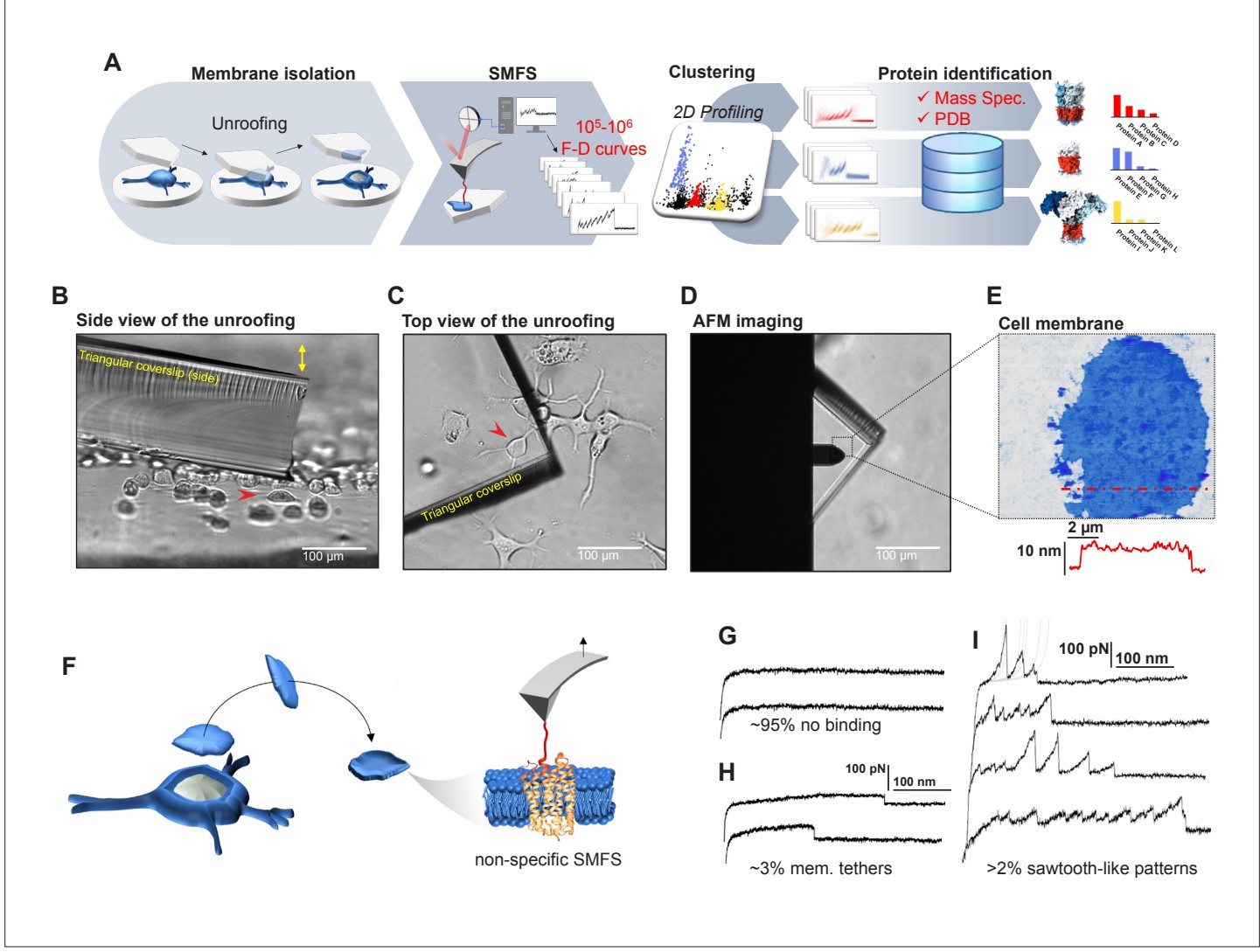

**Figure 1.** Experimental method for single-cell membrane isolation and protein unfolding. (**A**) Workflow of the method in four steps: isolation of the apical membrane of single cells; atomic force microscopy (AFM)-based protein unfolding of native membrane proteins; identification of the persistent patterns of unfolding and characterization of the population of unfolding curves; Bayesian protein identification thanks to mass spectrometry data, Uniprot and PDB. (**B**) Side view and (**C**) top view of the cell culture and the triangular coverslip approaching the target cell (red arrow) to be unroofed. (**D**) Positioning of the AFM tip in the region of unroofing. (**E**) AFM topography of the isolated cell membrane with profile. (**F**) Cartoon of the process that leads to single-molecule force spectroscopy (SMFS) on native membranes. Examples of force-distance (F-D) curves of (**G**) no binding events; (**H**) constant viscous force produced by membrane tethers during retraction; (**I**) sawtooth-like patterns, typical sign of the unfolding of a membrane protein.

The online version of this article includes the following figure supplement(s) for figure 1:

**Figure supplement 1.** Schematic setup for single-cell unroofing.

**Figure supplement 2.** In NG108-15 cells, membrane proteins remain into the membrane after unroofing, while cytoplasmic proteins do not.

**Figure supplement 3.** Absence of cytoplasmic proteins after unroofing of NG108-15 cells.

**Figure supplement 4.** Absence of cytoplasmic proteins in unroofed membranes in hippocampal cells.

**Figure supplement 5.** Absence of poly-D-lysine contamination in the single-molecule force spectroscopy (SMFS) data from unroofed cell membranes.

**Figure supplement 6.** Determination of the optimal persistence length of the detachment peak used for the total contour length calculation ($L_{cmax}$).

**Figure supplement 7.** Membrane proteins architectures.

**Figure supplement 8.** Candidates of multiple unfolding and origin of persistence length deviation.

**Figure supplement 9.** Clustering.

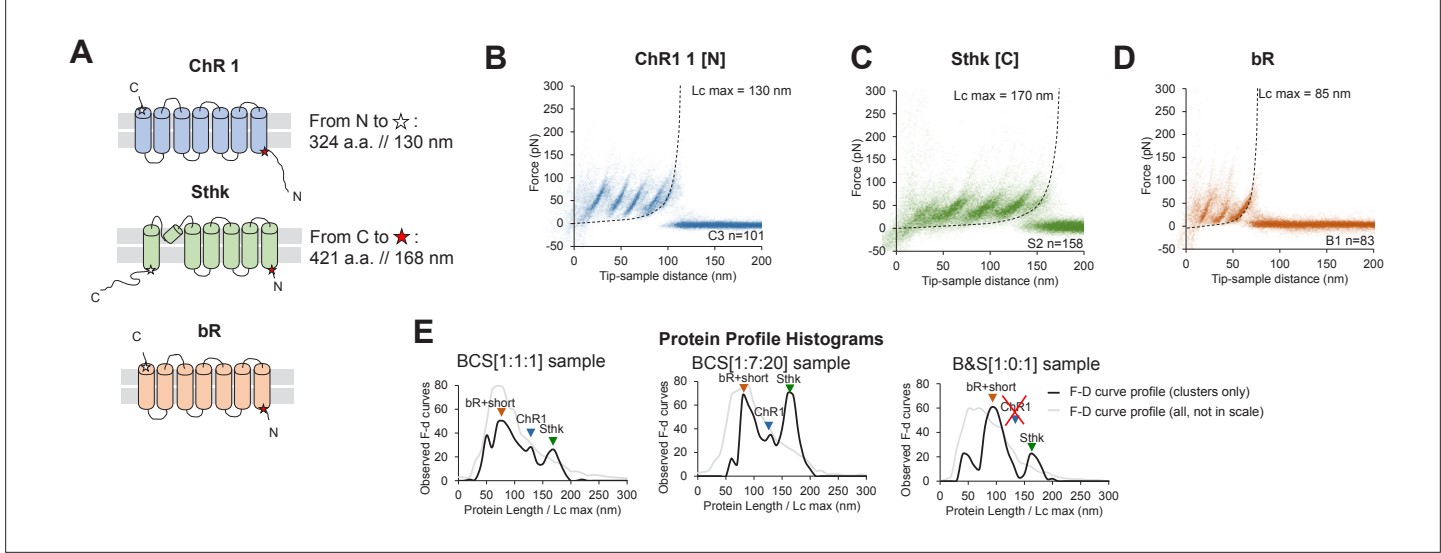

**Figure 2.** Unfolding of reconstituted mixtures of membrane proteins. (**A**) Scheme of the structure of the three purified proteins used in the in vitro preliminary step: Channelrhodopsin (ChR1), SthK, and Bacteriorhodopsin (bR) (cylinders represent α-helices). (**B**) Superimposition of 101 unfolding curves (density plot) of the full unfolding of ChR1 from the N-terminus. (**C**) Density plot of the full unfolding of Sthk from the C-terminus. (**D**) Density plot of the full unfolding of bR. (**E**) Protein profile, that is, the histogram of the maximal contour length ($L_{cmax}$) of all the force-distance (F-D) curves in the clusters (black line), and of all the F-D curves collected from the sample (gray line, not in scale), from samples with mixtures of bR, ChR1, and SthK with a relative abundance of 1:1:1, 1:7:20, and 1:0:1 (bR-SthK only). Arrows indicate where the F-D curves of (**B**), (**C**), and (**D**) accumulate in the histogram. The peak at ~90 nm indicated by bR+short also contains the shorter clusters of ChR1 and SthK shown in *Figure 2—figure supplement 1*, which also appear in the mixtures.

The online version of this article includes the following figure supplement(s) for figure 2:

**Figure supplement 1.** Unfolding of reconstituted membrane proteins.

**Figure supplement 2.** In vitro experiments with co-reconstitution of SthK and Channelrhodopsin (ChR1) (relative concentration 1:1) showing the five clusters obtained, and comparison of these new patterns S&Cx to the previous Cx and Sx clusters from single reconstitution.

**Figure supplement 3.** Clusters obtained in in vitro experiments with mixtures of Bacteriorhodopsin (bR), Channelrhodopsin (ChR1), and Sthk at different relative concentrations [1:1:1] and [1:7:20].

Therefore, there is no doubt that a large fraction of F-D curves will not represent the proper unfolding of a single protein. But inspired by the successes of single particle cryo electron microscopy (cryo-EM), which can produce atomic resolution structures, despite selecting less than 20% of the protein images (*Yi et al., 2019*), we attempted a similar approach on SMFS data, under the same hypothesis that 'bad' events are likely to produce *non-recurrent* patterns of unfolding.

The key tool to recognize recurrent patterns is an automated clustering method developed in our group (*Ilieva et al., 2020*) based on density peak clustering (*Rodriguez and Laio, 2014*; *Figure 1— figure supplement 9*, see Methods for a detailed description of the algorithm). In short, this approach detects statistically dense F-D patterns occurring often in the sample (clusters). It is based on a previously established 'similarity distance' (*Marsico et al., 2007*) in the context of SMFS and, remarkably, it does not require to pre-set neither the shape nor the number of clusters.

## Benchmarking the analysis with mixtures of purified proteins reconstituted in lipid bilayers

We first tested the consistency of our pipeline with three known membrane proteins. We reconstituted in lipid bilayers (*Shen et al., 2013*) highly purified Channelrhodopsin (ChR1) (*Nagel et al., 2002*), Bacteriorhodopsin (bR) (*Oesterhelt et al., 2000*), and the cyclic adenosine monophosphate gated channel SthK (*Marchesi et al., 2018*; *Figure 2A*), and we performed SMFS experiments (see *Supplementary file 2* for sample statistics). bR, ChR1, and SthK were reconstituted in vitro one at a time. Since we did not have control of the protein orientation in the bilayer, we expected two main F-D patterns, and therefore two clusters for each sample: one representing the unfolding of the protein from the N-terminus and the other from the C-terminus.

The clustering algorithm identified two almost indistinguishable clusters for bR (see Methods), and the result is in agreement with the fact that the unfolding of bR from the two termini is almost symmetric as previously reported (*Kessler and Gaub, 2006*; *Figure 2D*). In the case of ChR1 we obtained three clusters where the second cluster – cluster C2 – represents the partial unfolding of ChR1 from the C terminus because the peaks of C2 match perfectly the first three peaks of the full unfolding (cluster C1), while cluster C3 perfectly match the unfolding from the N-terminus (*Figure 2B*). Sthk generated two clusters, clusters S1 and S2 (*Figure 2C*), as expected (all the clusters are shown in *Figure 2—figure supplement 1*). For all the three proteins, we observed that: (i) the value of $L_{cmax}$ – that is, the estimated length of the segment of a.a. stretched in our SMFS experiments – is within 5% equal to the total length of the protein minus the free length of the N-terminus (C-terminus) domain if the cantilever tip attached to the C-terminus (N-terminus) (see *Figure 2B and C* and *Figure 2—figure supplement 1B*) and (ii) the force peaks of the clusters of F-D curves colocalize with the unfolded regions or loops of their structures (see *Figure 2—figure supplement 1A and C*).

Next, we simultaneously reconstituted two proteins in lipid bilayers (see Methods) obtaining the same clusters associated with the unfolding patterns observed in the single protein reconstitution (*Figure 2—figure supplement 2*). We finally mixed all the three proteins together, with different relative abundances: 1:1:1 and 1:7:20 respectively for bR, ChR1, and SthK. We observed that the number of traces per cluster scaled approximately with the abundance of the mix (*Figure 2—figure supplement 3*).

The histogram of the maximal contour length ($L_{cmax}$) of all the F-D curves (*Figure 2E*) encodes the protein content of the sample and allows a rapid detection of the presence and of the abundance of a given protein in sub-femtogram samples, also providing a qualitative indication on the concentration of a given protein. The success of this preliminary in vitro investigation prompted us to move to native membranes.

## Clustering SMFS data from native membrane proteins

Next, we applied our pipeline to DRG, hippocampal neurons, neuroblastoma, rod outer segments, and rod discs membranes (*Figure 3A–E*) and we obtained 301,654 curves from the hippocampal neurons, 413,468 from DRG neurons, 394,118 from neuroblastoma, 386,128 from rods, and 106,528 from rod discs. From these F-D traces we applied our clustering procedure that identified 15, 10, 11, 8, and 5 statistically dense pattern of unfolding present in the dataset (*Figure 3F–J*, the clusters are named with the sample identifier and a unique number for the cluster, for example, H3 refers to the third cluster from hippocampal cell membranes). These traces show that clean unfolding patterns can be detected even in patches of native membranes.

We identified four major classes of clusters: *Short curves with increasing forces:* clusters DRG12, H5, H8, and R3 show repeated peaks ($\Delta L_c$ 10–20 nm, distance between consecutive peaks) of increasing force up to 400 pN. *Long and periodic curves:* R6, H7, and DRG10 display periodic peaks of ~100 pN and with a $\Delta L_c$ of 30–40 nm whose unfolding patterns are similar to what seen when unfolding LacY (*Serdiuk et al., 2016*). *Conventional short curves:* the majority of the identified clusters like DRG1, H3, R8, and all clusters from the rod discs have F-D curves with a total length less than 120 nm with constant or descending force peaks. These F-D curves share various features with the opsin family unfolded in purified conditions (*Engel and Gaub, 2008*), for example, a conserved unfolding peak at the beginning (at contour length <20 nm) associated to the initiation of the denaturation of the protein. We also found *unconventional* clusters such as DRG7, DRG8, and R7: DRG8 exhibits initial high forces and with variable peaks followed by more periodic low forces, while cluster R7 has a conserved flat plateau at the end of the curve of unknown origin.

A compact representation of clustered F-D curves becomes more important in native samples where the information stored in the SMFS data is more complex (see *Figure 3L–P*). The cancerous NG108-15 cells have few short and not particularly stable membrane proteins compared to neural cells which instead have many clusters of proteins under 100 nm that unfold even above 200 pN. On the contrary, NG108-15 cells have a higher fraction of long and stable proteins than neurons.

Furthermore, in native membranes we can directly compare the expected protein profile obtained from mass spectrometry (usually obtained from thousands of cells, see Methods) with the F-D curves profile of just few cells (*Figure 3K*) which shows a good agreement in particular in the region above 100 nm of protein length.

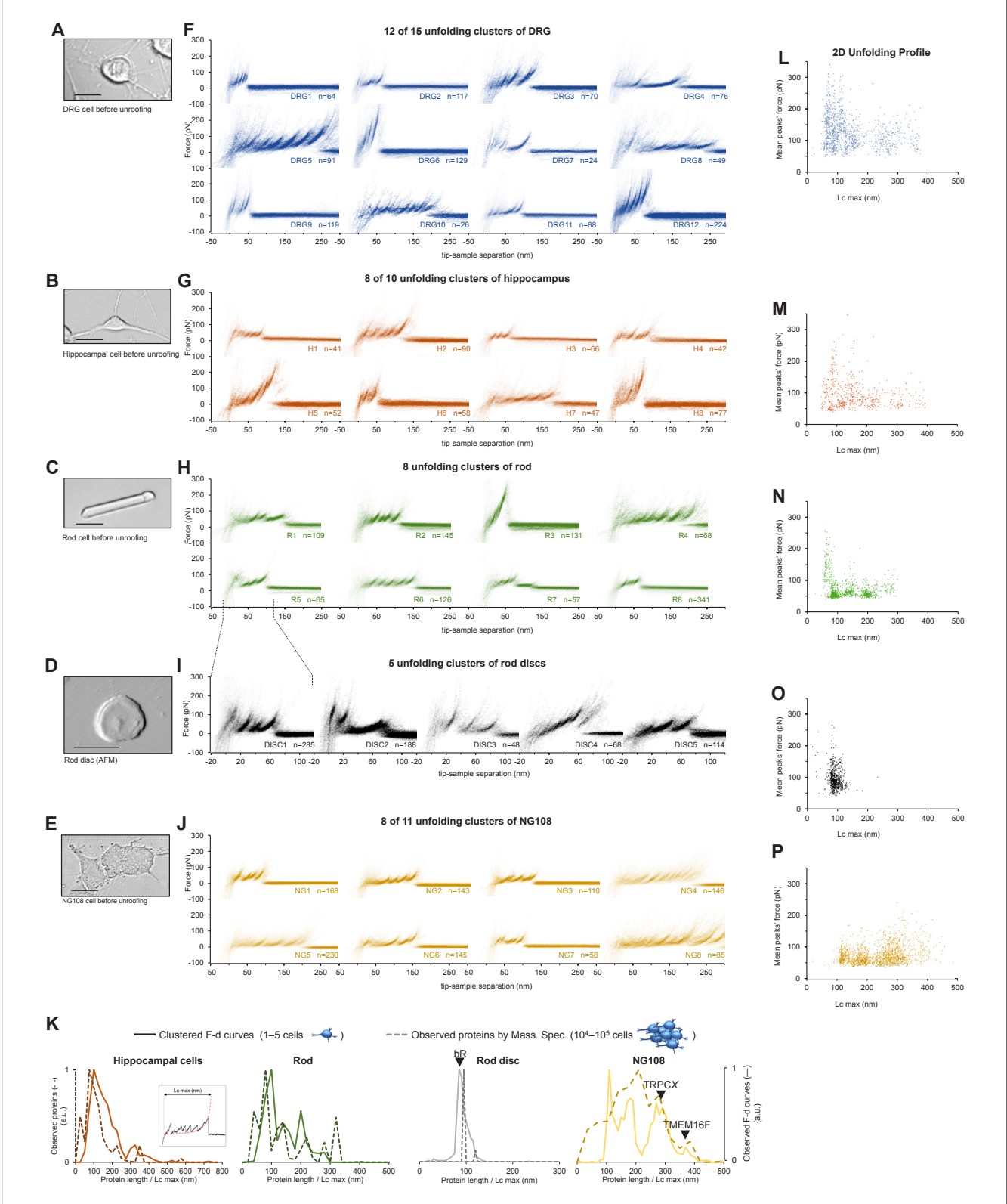

**Figure 3.** Unfolding clusters in native cell membranes. Bright-field image of (**A**) dorsal root ganglia neuron; (**B**) hippocampal neuron; (**C**) rod before unroofing (scale bar 15 μm). (**D**) Atomic force microscopy (AFM) error image of an isolated disc (scale bar 1 μm), (**E**) NG108-15 cells. (**F, G, H, I, J**) Examples of obtained clusters from the native membranes shown as density plots, that is, superposition of the *n* unfolding curves agglomerated and colored as a heatmap. (**K**) Comparison of the protein profiles obtained with mass spectrometry vs. the force-distance (F-D) curves obtained with single-

*Figure 3 continued on next page*

*Figure 3 continued*

molecule force spectroscopy (SMFS) shows a good correlation. The protein profile from mass spectrometry is the normalized histogram of the total number of proteins found in the sample, considering their abundance. The protein profile from SMFS is the histogram of the maximal contour length ($L_{cmax}$) of all the F-D curves selected with our clustering procedure. (**L, M, N, O, P**) Representation of all the clustered F-D curves in 2D: x-axis is the maximum contour length; y-axis is the average unfolding force (DRG: $n$=1255; hippocampus: $n$=563; rod: $n$=1039; disc: $n$=703, NG108-15 cells: $n$=1591).

The online version of this article includes the following figure supplement(s) for figure 3:

**Figure supplement 1.** Alternative visualizations of cluster analysis.

## Molecular identification of detected clusters

Having identified clusters of F-D curves from native membranes that approximate the overall population of membrane proteins (*Figure 3K*), the next question is: can we identify the membrane proteins whose unfolding corresponds to the identified clusters in *Figure 3*?

For this purpose, we developed a Bayesian method providing candidate proteins on the basis of the information present in the data from mass spectrometry (ProteomeXchange) and in other proteomic databases (Uniprot, PDB) – combined with the empirical results obtained from SMFS literature (*Bosshart et al., 2012*; *Cisneros et al., 2005*; *Ge et al., 2016*; *Kawamura et al., 2013*; *Kedrov et al., 2004*; *Klyszejko et al., 2008*; *Maity et al., 2015*; *Möller et al., 2003*; *Oesterhelt et al., 2000*;

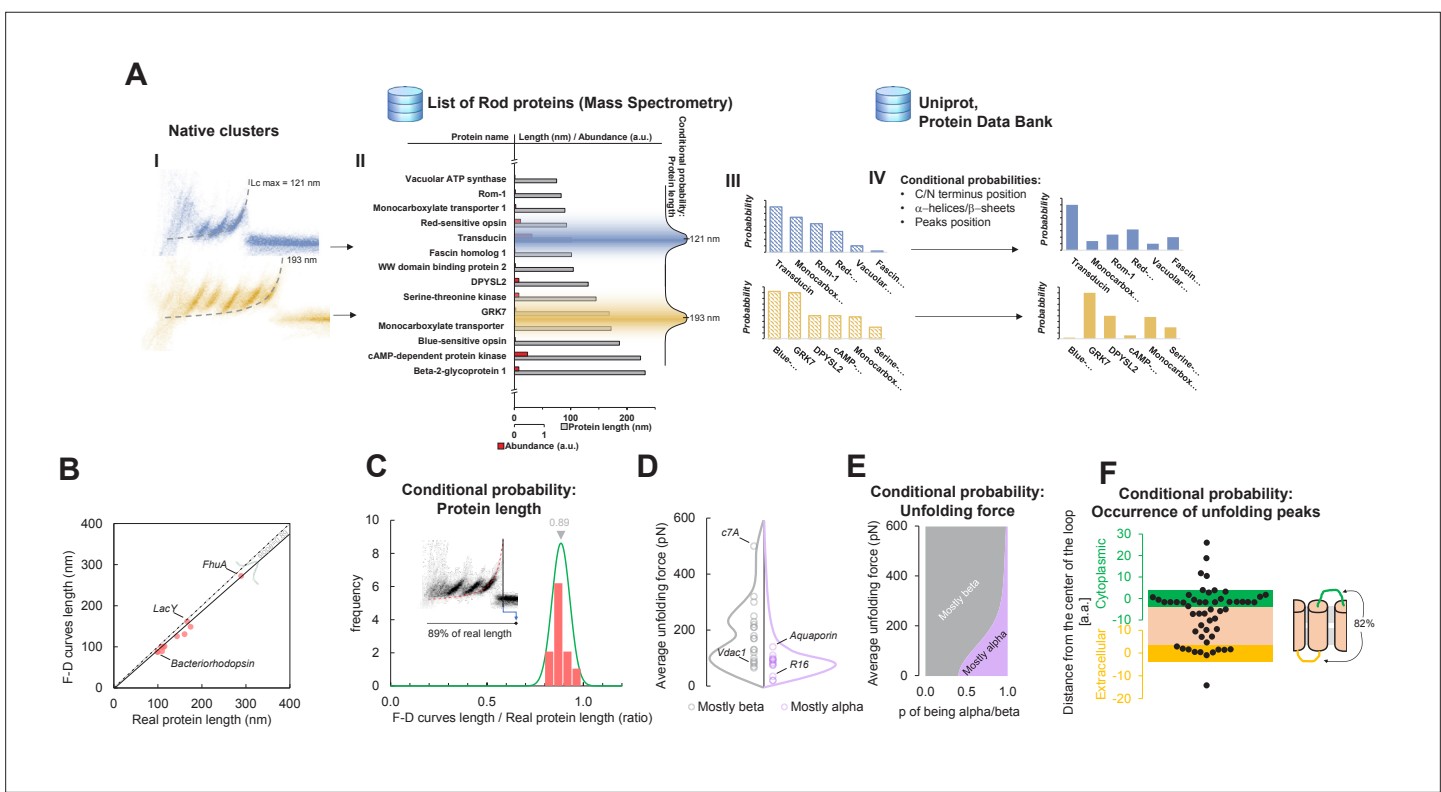

**Figure 4.** Bayesian identification and conditional probabilities. (**A**) Logical workflow of the Bayesian steps: selection due to total length and abundance (mass spectrometry), refinement with structural and topological information (PDB and Uniprot). (**B**) Comparison of the real length of the protein vs. the measured maximal contour length of the force-distance (F-D) curves in 14 single-molecule force spectroscopy (SMFS) experiments on membrane proteins (Pearson coefficient $r$=0.991). (**C**) Conditional probability of the observed maximal length of the clusters obtained from (**B**). (**D**) Comparison of the force necessary to unfold β-sheets and α-helices in 22 SMFS experiments. (**E**) Conditional probability of the observed unfolding forces obtained from (**D**). (**F**) Conditional probability for the occurrence of unfolding peaks extracted from SMFS literature (see Methods). Unfolding peaks occurs most likely in the loops (82%) than in transmembrane domains (n_peaks = 54, from 11 SMFS experiments of different membrane proteins). The points in the green (yellow) region represents unfolding peaks occurred in a cytoplasmic (extracellular) loop, the point in the pink area occurred in a transmembrane domain. The points above the green and below the yellow regions occurred in cytoplasmic and extracellular domains, respectively. The scale is approximate because in rare occasion loops are longer than 10 nm.

The online version of this article includes the following figure supplement(s) for figure 4:

**Figure supplement 1.** The cross-correlation used to evaluate $P\left(Peaks_{Cx}Structure_{Prot_A}\right)$.

*Sapra et al., 2009*; *Serdiuk et al., 2016*; *Thoma et al., 2017*; *Thoma et al., 2012*). The Bayesian identification (*Figure 4A*) is based on two steps: first, the crossing of information between the clusters with the results of mass spectrometry analysis of the sample under investigation (hippocampal neurons, discs, etc.); second, a refinement of the preliminary candidates using additional information (structural and topological) present in the PDB and Uniprot databases. The first step relies on the comparison of $L_{cmax}$ of each cluster with the length of proteins identified in the mass spectrometry data (*Chen et al., 2006*; *Kwok et al., 2008*; *Panfoli et al., 2008*). By using the protein abundance included in the mass spectrometry data as the prior, a first list of candidate proteins is obtained as well as their probabilities (*Figure 4A* III). The SMFS literature contains 14 examples of unfolded membrane proteins allowing a comparison between the $L_{cmax}$ of the measured F-D curves and the real length of the same protein completely stretched (*Figure 4B*). Therefore, we extrapolated the first conditional probability of our Bayesian inference, showing that on average, the value of $L_{cmax}$ corresponds to 89% of the real length of the protein ($R^2$=0.98). When the structure of candidate proteins is known (helices, sheets), these probabilities are further refined considering for example, the force of unfolding or the position of the loops which are known to constrain the possible unfolding patterns (*Figure 4C–F*, see Methods for the formal description).

Like in mass spectrometry, this identification is probabilistic (*Figure 5*), and it allows to reduce the number of possible candidates from hundreds to one to four proteins.

To prove the reliability of the proposed pipeline protocol, we focused on four clusters found in NG108-15 cells where the Bayesian method assigned a high probability to the membrane proteins TMEM16F, TRPC1, TRPC5, and TRPC6. Reverse transcription polymerase chain reaction and western blot analysis (*Wu et al., 2007*; *Figure 6—figure supplement 1*) confirm that these membrane proteins are abundantly expressed in neuroblastoma NG108-15 cells. We overexpressed the construct 6xHis-N2B-*protein*-GFP, where *protein* is the molecular candidate to be validated (*Figure 6A–B*, *Supplementary file 1*). The N2B is a chain of 204 aa which can be unfolded with a force less than 10 pN providing a well-known signature (*Linke and Grützner, 2008*), and GFP is the green fluorescent protein used to identify which NG108-15 cell was successfully transfected, and if the proteins were present in the unroofed membrane patch (*Figure 6C*).

We compared the clusters obtained without overexpression (NG8, NG4, NG5, and NG11) with the new clusters obtained from the unfolding of the constructs N2B-TMEM16F-GFP, N2B-TRPC1-GFP, N2B-TRCP5-GFP, and N2B-TRPC6-GFP from the transfected cells (*Figure 6D and E*). These F-D traces had the well-known signature of the N2B domain consisting in a flat portion with a length of ~85 nm, which was followed by force peaks almost exactly matching those observed in the corresponding density plots (*Figure 6D*). This matching was also confirmed by the correspondence of the cumulative peaks observed in the global histogram of the contour length of the NG8, NG4, NG5, and NG11 clusters with those obtained from neuroblastoma cells transfected with the constructs N2B-TMEM16F-GFP, N2B-TRPC1-GFP, N2B-TRCP5-GFP, and N2B-TRPC6-GFP translated of ~85 nm because of the presence of the N2B (*Figure 6E*).

This pipeline demonstrates that from the obtained clusters (*Figure 3*) it is possible to derive the unfolding spectra of membrane proteins in their almost-native conditions without the need of purification. The superposition of the peaks (*Figure 6D–E*) confirms the identification and allows to map the unfolding positions along the tertiary structure of the proteins (*Figure 6F*).

## Structural insights from SMFS

The confirmation of the molecular identity of the unfolded proteins allows a better understanding of their molecular structures obtained from cryo-EM, in particular of the not well-folded regions. SMFS detects the unfolded domains of a protein and can characterize their variability where the more powerful cryo-EM precisely determines the position of the atoms of the well-folded domains. The unfolding of the construct N2B-TRPC6-GFP as well as the endogenous TRPC6 show force peaks at 143, 193, and 230 nm with different properties (*Figure 7*). The peak at 143 nm is always well defined, but the force peaks at 193 and 230 nm – occurring at the extracellular loop E1 between S1 and S2 and at the extracellular loop E2 between S3 and S4, respectively (see positions in *Figure 7B*) – show variability in the location and strength of the secondary peaks (*Figure 7D and E*). The unfolding behavior observed at 193 and 230 nm reflects the presence of small structural elements (*Yu et al., 2017*) which are present in some but not all instances (see inserts in *Figure 7D–E*). The peak at 143 nm in contrast

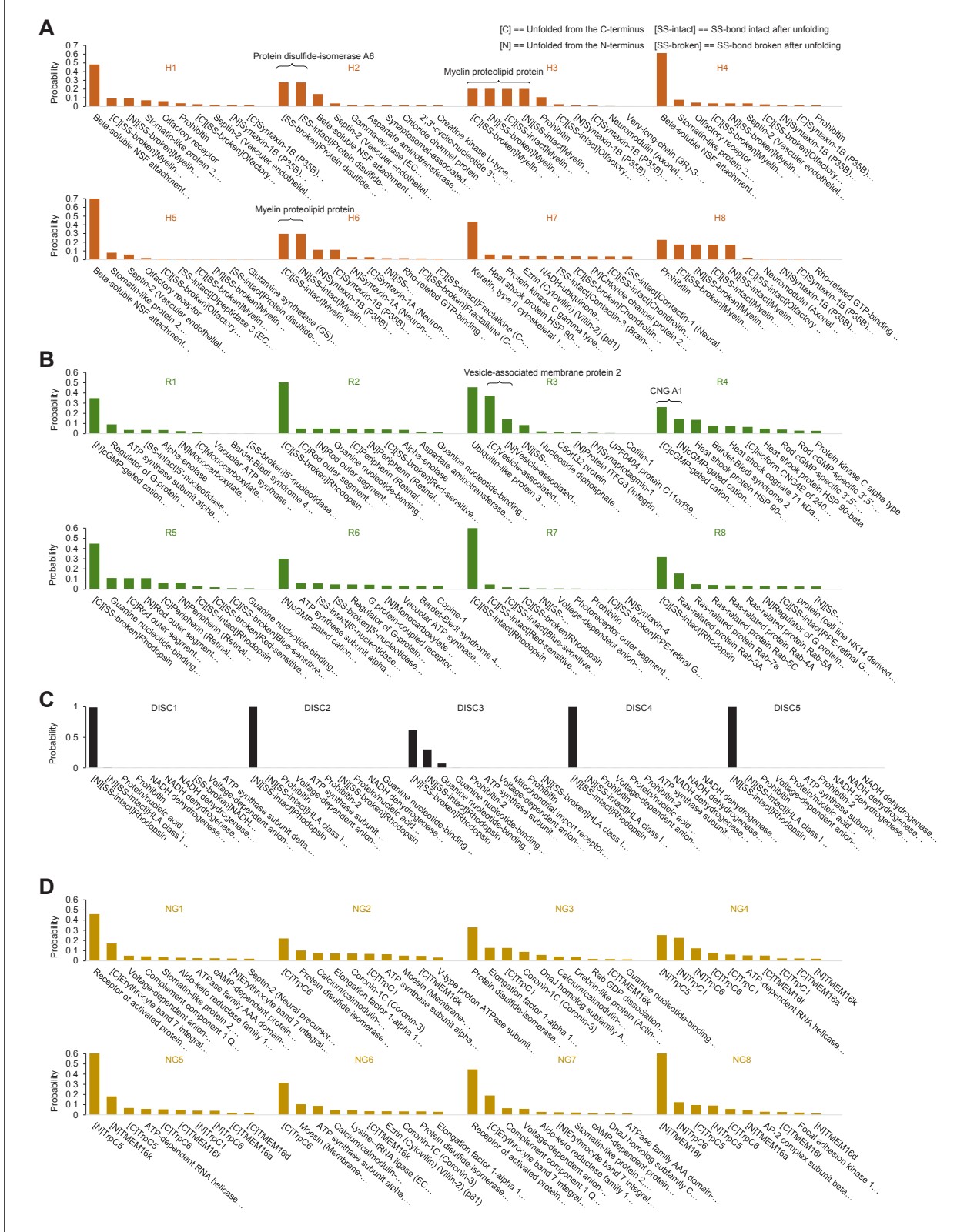

**Figure 5.** Bayesian identification of the unfolding clusters. Most probable candidates for the unfolding clusters found in (**A**) hippocampal neurons; (**B**) rods; (**C**) rod discs; (**D**) NG108-15 cells.

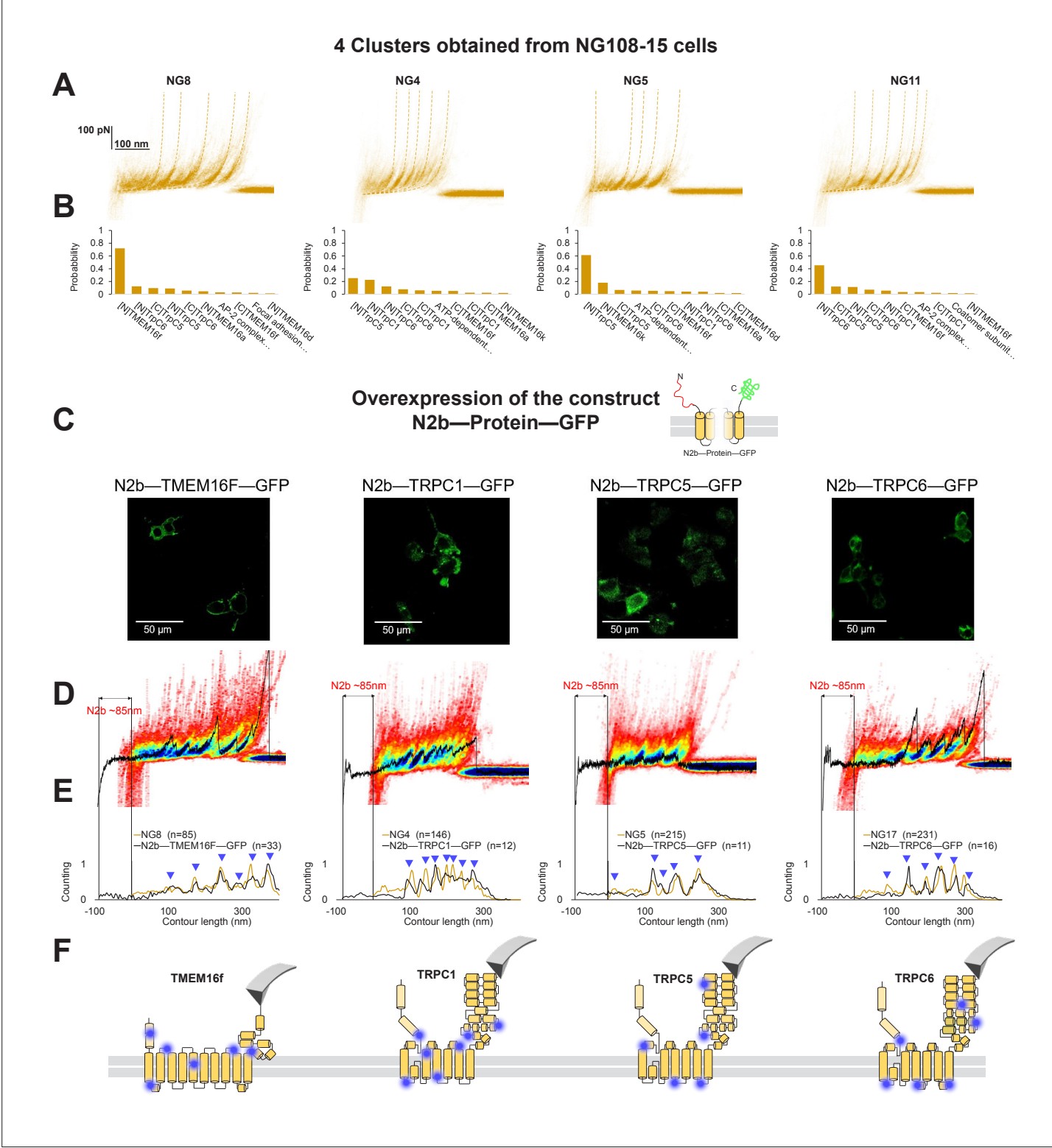

**Figure 6.** Validation of the method with a comparison between native clusters and those obtained with overexpression of a construct with the N2B signature at the N-terminal and GFP at the C-terminal. (**A**) Four clusters found in the dataset of force-distance (F-D) traces pulled from NG108-15 cells wild type. (**B**) The Bayesian identification of the clusters in (**A**) with the candidate membrane proteins. (**C**) Confocal images of NG108-15 cells overexpressing the construct N2b-Protein-GFP, as reported by the green fluorescence emitted by GFP. (**D**) From left to right, color density plots (blue indicating a colocalization among F-D traces of more than 80%) of the clusters superimposed to an F-D curve bearing the N2B signature

*Figure 6 continued on next page*

*Figure 6 continued*

(85 nm segment with a flat force below 10 pN) obtained in the sample overexpressing the candidate protein TMEM16F, TRPC1, TRPC5, and TRPC6, respectively. (**E**) Global histogram of $L_c$ of the clusters in A and the clusters with the N2B signature from cells transfected with the corresponding construct. (**F**) Position of the most likely rupture regions of the proteins according to (**E**) superimposed to the cartoon of the cryo electron microscopy (cryo-EM) structure available in the PDB.

The online version of this article includes the following source data and figure supplement(s) for figure 6:

**Figure supplement 1.** Western blots for the four proteins indicating the silencing obtained with different plasmids.

**Figure supplement 1—source data 1.** Photos of the western blots.

shows a clear and reproducible unfolding behavior (*Figure 7C*) typical of a defined and well-folded structure (*Rico et al., 2013*; *Takahashi et al., 2018*).

In the light of these results, we analyzed the available cryo-EM structures of TRPC6. There are three published structures (*Bai et al., 2020*; *Tang et al., 2018*); these three cryo-EM structures are

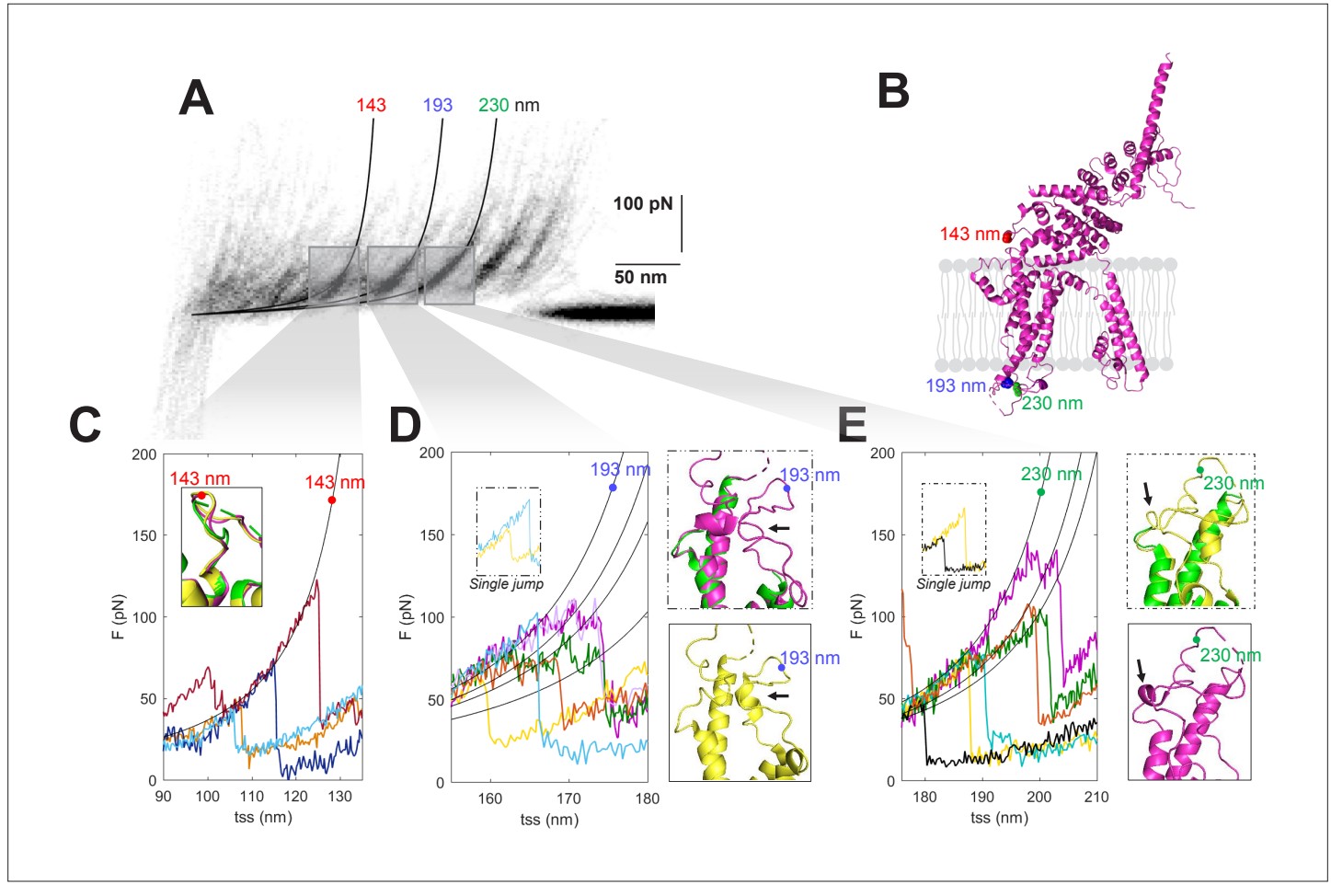

**Figure 7.** Structural segmentation in the loops of TRPC6. (**A**) Density plot of force-distance (F-D) curves of TRPC6 (validated from cluster NG11). (**B**) 3D structure of TRPC6 (PDB: 6UZA) with highlighted the position corresponding to the unfolding peaks in (**A**). (**C**) Representative F-D curves showing the single peak at 143 nm. Insert: comparison of the agreement with the three proposed structures of TRPC6 (PDB ID, green: 5YX9; purple: 6UZA; yellow: 6UZ8). (**D**) Representative F-D curves showing the variable behavior at 193 nm where ~20% of the F-D curves shows a single jump that suggests no structural fragmentation of the loop like in the green/purple structures. Right inserts: comparison of the structure of the loop obtained by cryo electron microscopy (cryo-EM) where only the yellow structure shows the presence of a 2-turn helix. (**E**) Representative F-D curves at 230 nm where ~30% of the F-D curves shows a single jump that suggests no structural fragmentation of the loop like in the green/yellow structures. Right inserts: comparison of the structure of the loop obtained by cryo-EM where only the purple structure shows the presence of a 1-turn helix.

The online version of this article includes the following figure supplement(s) for figure 7:

**Figure supplement 1.** Unfolding intermediates and humps.

very similar – and almost identical – particularly in the long transmembrane α-helices, but differ in the connecting loops where short α-helices are present only in some of these structures (*Figure 7C–E*, inserts). These short α-helices in the loops E1 and E2 are the small structural elements at the origin of the segmentation observed in the F-D traces (*Figure 7D–E*). However, as shown in the structures in yellow, the α-helices are not always observed as the single-peak F-D curves suggest. Therefore, the most likely conclusion from our SMFS data is that, at RT and in their native environment, the different cryo-EM structures coexist and the connecting loops E1 and E2 have a variable structure.

The observed variability is not restricted only to loop E1 and E2 of TRPC6: also the loop pre-S1 and S1 of TRPC5 shows a similarly complex unfolding behavior as opposed to the unfolding of the loop between S2 and S3. Clusters DRG5 or H3 show an even more exotic behavior. Their unfolding curves show a dual nature: secondary peaks and unfolding humps (*Figure 7—figure supplement 1*).

### Towards membrane proteomics on single cells

One of the most important results of this work are the protein profiles of *Figure 2E* and *Figure 3K*. SMFS experimentalists are aware that the collected dataset are very heterogeneous even with highly pure samples. And this is indeed reflected in the gray profiles of *Figure 2E* where no clear data structure is present, and where the three samples are indistinguishable from each other. But after the clustering procedure described in this manuscript, that requires minimal interventions by the user, we could obtain the black profiles in *Figure 2E* which clearly show the presence of structures in the data that have a straightforward correspondence with the samples. This protein profile can be considered as the minimal 2D representation of the SMFS dataset, but more importantly a 'first-hand information' on the mass/length of the proteins in the sample. This positive outcome encouraged us to analyze the SMFS dataset from native cells in the same way. *Figure 3K* shows the comparison of the distribution of membrane proteins obtained with mass spectrometry (broken lines) against the distribution of the F-D curves that survived the clustering protocol (solid lines). The solid lines show broader peaks and the distributions are not matching perfectly, but we think that the similarity is still remarkable considering the several orders of magnitude difference on the amount of sample used by the two techniques. To become quantitatively reliable, this approach still needs appropriate improvements, but these results suggest that protein length detection of SMFS – restricted to clustered traces – provides a semi-quantitative proxy for mass detection at the single-cell membrane level. These results may pave the way to single-cell proteomics and phenotyping, with the potential to advance our understanding of disease development, progress, and treatment effects.

## Discussion

Given the high diversity of membrane proteins in cells, the unequivocal identification of unlabeled proteins from single cells is a daunting task. The present manuscript shows that by combining SMFS on native membrane patches with information present in databases, a probabilistic identification of groups of proteins is possible. Our approach provides usually less than four candidates (*Figure 5*) for most of the identified clusters of F-D traces (p>95%), and we were able to prove the validity of the identification by properly engineered constructs (*Figure 6*).

### Advantages of the method

The proposed pipeline offers the possibility to enrich the structural information usually derived from cryo-EM, with insights obtained by unfolding the proteins in their native environment. SMFS cannot distinguish between a β-sheet and an α-helix, but it can determine with a good accuracy the unfolded domains of a membrane protein at RT (see *Figure 7*; *Engel and Gaub, 2008*). Cryo-EM cannot capture the structural heterogeneity of the poorly folded regions, while SMFS can provide important complementary information, for instance about the mechanical stability of connecting loops.

The pipeline, in addition, offers the possibility to obtain the unfolding profile of the membrane proteins from a limited amount of native material (membranes isolated from 1 to 10 cells). We have shown that the unfolding profile of membrane proteins can be used as a fingerprint to characterize the sample under investigation (see *Figures 2–3* and *Figure 3—figure supplement 1*). We foresee that this approach can be extended to characterize, and eventually distinguish, membranes in cells in

healthy and sick conditions where other methodology cannot be applied because of sample scarcity (e.g. samples from patients).

## General hallmarks of SMFS of membrane proteins

The unfolding of a protein is stochastic in nature because it can be viewed as Brownian diffusion of a particle in a tilted energy landscape (*Engel and Gaub, 2008*; *Yu et al., 2017*), therefore variability in F-D curves is intrinsic to the technique. But this variability is not unconstrained: the energy landscape of a membrane protein has obligatory intermediates that allow to cluster the F-D curves coming from the same protein (*Marsico et al., 2007*). The variation of an obligatory intermediate, or, in other words, a new force peak/a shift of a force peak, is a strong evidence of a conformational change in the system (*Maity et al., 2015*). Minor unfolding features can also provide evidence of mechanical plasticity (*Takahashi et al., 2018*) or structural segmentations of apparently continuous structures (*Yu et al., 2017*).

Here, we want to stress that we are referring to the 'system' *protein* plus *the surrounding environment* and not to the *protein* only. The result of a pulling experiment of a membrane protein, as opposed to globular proteins, is strictly speaking not a single biomolecule event. A membrane protein does not exist in isolation because at least it needs lipids, but very often it forms oligomers with other proteins or even more complex structures like G-protein-coupled receptors. Therefore, to study the mechanical stability of a membrane protein it is preferable to have it in its physiological environment, whether it is composed of only lipids or other partner proteins that concur in the stabilization of the structure (like in the CNG tetramer; *Maity et al., 2015*; *Napolitano et al., 2021*).

## Technical limitations of the method and future directions

A major limitation of the proposed pipeline – in its present form – is the possibility to merge in the same cluster unfolding traces of proteins with a different molecular identity: indeed, from the mass spectrometry data it is clear that different proteins can have the same – or approximately the same – molecular weight, a similar unfolding pattern and a similar total unfolded length $L_c$. This issue is rather significant for short proteins, that is, those with values of $L_c$ between 50 and 150 nm. In order to overcome this limitation, it will be desirable to couple SMFS with a device (e.g. solid-state nanopores; *Lee et al., 2018*) able to identify, even qualitatively, specific amino acids during the unfolding process.

In the range between 50 and 150 nm, we also expect a number of F-D curves resulting from arbitrary attachment or detachment to pass the quality filter (see Methods Block 2). But in this case, given their arbitrariness, the similarity distance will position them at the edge of dense clusters so they will not be central in the analysis.

An underlying technical problem is that it is not trivial to assess a priori the expected maximal contour length (max $L_c$) simply by knowing the length of the protein measured by mass spectrometry (which usually corresponds to the nominal length: Number of amino acids * 0.4 nm). The difference between the two lengths depends on the specific anchoring points of the protein: the measured max $L_c$ corresponds to the length of the chain between the tip of the AFM and the last amino acid anchored to the membrane during the pulling. In our experiments on reconstituted proteins (see *Figure 2A–D*), and in many reports from the literature (*Kawamura et al., 2013*; *Kessler and Gaub, 2006*; *Möller et al., 2003*), the last anchored point is the last amino acid of the last transmembrane segment (within error). However, applying these heuristics to all the proteins present in the mass spectrometry dataset is not feasible because the structure of many of those proteins is not known, and even if known, the error on this estimate is difficult to be assessed. We decided to use a Bayesian framework that is more flexible and that naturally provides an error estimate of the results. In our framework we rescaled the nominal length with the probability distribution shown in *Figure 4C*. Based on the 14 references found in the literature, the max $L_c$ is equal to 89% of the length measured by mass spectroscopy on average. The Pearson correlation coefficient is $r=0.991$, reflecting a very good correlation between the two quantities (see *Figure 4B*). Therefore, we think that this approach is more scalable, and easier to be updated when new data will become available from the literature.

The Bayesian approach we used is also robust in case of a large conformational change of a protein as long as the last anchoring point does not change more than 10% of the max $L_c$. The max $L_c$ has the higher weight in the probability estimation while the contribution of the correlation of the peaks (see

*Figure 4—figure supplement 1*) is a second-order correction. This means that the same protein with different conformation is still properly classified.

Abundancy of the membrane proteins in the isolated membrane is also a limiting factor: the number of distinct membrane proteins that are present, for example, in hippocampal neurons is believed to be in the order of hundreds but with only few of them representing the majority of the expressed proteins. With our approach indeed we identify only 10 clusters that are then assigned to some of the most abundant proteins in the respective contour length range (see Methods for a first-order assessment of the relation between the number of collected traces and the predicted dimension of a cluster).

A way to mitigate this issue would be to functionalize the cantilever tip to increase the yield of unfolding specific target proteins. The specific functionalization strategy really depends on the goal of the investigation. The goal of the present manuscript is to perform unbiased SMFS and we think that the relatively low yield is an advantage in our case. A high attachment rate would cause a higher probability of binding two proteins at the same time making the F-D curves analysis more complicated if not impossible to interpret. On the contrary, attachment rates of 2% means that in first approximation double binding is also the 2% of the binding events so the 0.04% overall: this makes the analysis less error-prone.

Finally, an inherent characteristic of the isolation technique here presented is that only the cytoplasmic side of the membrane is accessible, and a method to reverse the membrane in order to perform the experiments from the extracellular side would be desired. This would require a purely technical improvement. We can speculate thinking about a device with multiple pipettes with which to perform parallel suctions on the cell membrane, to break the membrane as in a patch clamp inside-out configuration and then release the membrane on a clean surface coated with poly-lysine. The proof of principle of such an improvement is not demonstrated yet.

This work is a first step for the use of SMFS in native samples, but despite the limitations here reported, we were still able to classify tens of unfolding pathways, identify with high accuracy part of them, and we could also reconcile some contrasting cryo-EM structures. We therefore believe that the proposed methodology represents a genuine step in the direction of single-cell membrane proteomics.

# Materials and methods

**Key resources table**

| Reagent type (species) or resource | Designation | Source or reference | Identifiers | Additional information |
|---|---|---|---|---|
| Gene (*Mus musculus*) | TMEM16F | PMID:34445284 | N/A | |
| Gene (*Mus musculus*) | TMEM16A | PMID:34089532 | N/A | |
| Gene (*Spirochaeta thermophila*) | SthK | PMID:30266906 | N/A | pCGFP-BC vector (modified by deleting the GFP and four out of the eight histidines) |
| Gene (*Mus musculus*) | Mouse TRPC1 | MiaoLing Plasmid Company (NM_011643.4) | N/A | |
| Gene (*Mus musculus*) | Mouse TRPC5 | MiaoLing Plasmid Company (NM_009428.3) | N/A | |
| Gene (*Mus musculus*) | Mouse TRPC6 | MiaoLing Plasmid Company (NM_013838.2) | N/A | |
| Gene (*Mus musculus*) | N2B | PMID:25963832 | N/A | |
| Strain, strain background (*Escherichia coli*) | DH5α | Thermo Fisher Scientific | Cat#18258012 | Competent cells |
| Strain, strain Background (*Escherichia coli*) | Stbl3 | Thermo Fisher Scientific | Cat#C7373-03 | Competent cells |
| Cell line (*Mouse × Rat Hybrid*) | NG108-15 cells | Sigma-Aldrich (China) | Cat#88112302 | |

*Continued on next page*

*Continued*

| Reagent type (species) or resource | Designation | Source or reference | Identifiers | Additional information |
|---|---|---|---|---|
| Transfected construct (*Mus musculus*) | TMEM16A-GFP | PMID:34089532 | N/A | peGFP-N1 backbone |
| Transfected construct (*Mus musculus*) | TMEM16F-GFP | PMID:34445284 | N/A | peGFP-N1 backbone |
| Transfected construct (*Mus musculus*) | 6xHis-N2B- mTMEM16F-GFP | GENEWIZ (China) | N/A | peGFP-N1 backbone |
| Transfected construct (*Mus musculus*) | 6xHis-N2B- mTRPC1-GFP | GENEWIZ (China) | N/A | peGFP-N1 backbone |
| Transfected construct (*Mus musculus*) | 6xHis-N2B- mTRPC5-GFP | GENEWIZ (China) | N/A | peGFP-N1 backbone |
| Transfected construct (*Mus musculus*) | 6xHis-N2B- mTRPC6-GFP | GENEWIZ (China) | N/A | peGFP-N1 backbone |
| Transfected construct (*Mus musculus*) | plentiCRISPR V2-TRPC1-sgRNA1 | MiaoLing Plasmid Company | N/A | sgRNA sequence ccgtaagcccacctgtaaga |
| Transfected construct (*Mus musculus*) | plentiCRISPR V2-TRPC1-sgRNA2 | MiaoLing Plasmid Company | N/A | sgRNA sequence acgcttgtagcagaagggct |
| Transfected construct (*Mus musculus*) | plentiCRISPR V2-TRPC5-sgRNA1 | MiaoLing Plasmid Company | N/A | sgRNA sequence attactctacgccatccgca |
| Transfected construct (*Mus musculus*) | plentiCRISPR V2-TRPC5-sgRNA2 | MiaoLing Plasmid Company | N/A | sgRNA sequence ggagtgtgtatccagttcgg |
| Transfected construct (*Mus musculus*) | plentiCRISPR V2-TRPC6-sgRNA1 | MiaoLing Plasmid Company | N/A | sgRNA sequence gcggcagacgattcttcgtg |
| Transfected construct (*Mus musculus*) | plentiCRISPR V2-TRPC6-sgRNA2 | MiaoLing Plasmid Company | N/A | sgRNA sequence taaaggttatgtacggattg |
| Transfected construct (*Mus musculus*) | plentiCRISPR V2-TMEM16F-sgRNA1 | MiaoLing Plasmid Company | N/A | sgRNA sequence agcgagcgttacctcctgta |
| Transfected construct (*Mus musculus*) | plentiCRISPR V2-TMEM16F-sgRNA2 | MiaoLing Plasmid Company | N/A | sgRNA sequence ctctcgggtcaaataccaag |
| Transfected construct (*Mus musculus*) | plentiCRISPR V2-control-sgRNA | MiaoLing Plasmid Company | N/A | sgRNA sequence tcttgagtttgtaacagctg |
| Transfected construct (*Mus musculus*) | mCherry-Lifeact-7 | Addgene | Plasmid #54491 | Actin labeled with mCherry |
| Biological sample (*Rat*) | Hippocampal and DRG neurons | Home-made | N/A | Freshly isolated from Wistar rats |
| Biological sample (*Xenopus laevis*) | Rod cells | Home-made | N/A | Freshly isolated from male *Xenopus laevis* |
| Antibody | Anti-TMEM16F (Rabbit polyclonal) | Alomone Labs | Cat#ACL-016 | WB(1:200) |
| Antibody | Anti-TRPC1 (Rabbit polyclonal) | Alomone Labs | Cat#ACC-010 | WB(1:500) |
| Antibody | Anti-TRPC5 (Rabbit polyclonal) | Alomone Labs | Cat#ACC-020 | WB(1:500) |
| Antibody | Anti-TRPC6 (Rabbit polyclonal) | Alomone Labs | Cat#ACC-120 | WB(1:500) |
| Antibody | Anti-α-Tubulin (Mouse monoclonal) | Sigma | Cat#T8203 | WB(1:5000) |
| Antibody | Goat Anti-Rabbit HRP (Goat polyclonal) | Dako | Cat#P0448 | WB(1:5000) |

*Continued*

| Reagent type (species) or resource | Designation | Source or reference | Identifiers | Additional information |
|---|---|---|---|---|
| Antibody | Goat Anti-Mouse HRP (Goat polyclonal) | Dako | Cat#P0447 | WB(1:5000) |
| Peptide, recombinant protein | SthK | Home-made | N/A | |
| Peptide, recombinant protein | Hs Bacteriorhodopsin (bR) | Cube-Biotech | Cat#28903 | |
| Peptide, recombinant protein | Channelrhodopsin 1_Ca (ChR1) | Cube-Biotech | Cat#28941 | |
| Chemical compound, drug | TCEP | Sigma | Cat#C4706 | |
| Software, algorithm | Matlab2017a | MathWorks | N/A | |
| Software, algorithm | ImageJ 1.47v | NIH | RRID:SCR_003070 | |
| Other | Hoechst | Life Technologies | Cat#33342 | Stain the DNA in live cells without the need of permeabilization |

All experimental procedures were in accordance with the guidelines of the Italian Animal Welfare Act, and their use was approved by the SISSA Ethics Committee board and the National Ministry of Health (Permit Number: 630-III/14) in accordance with the European Union guidelines for animal care (d.1.116/92; 86/609/C.E.).

## Cell preparation and culture
### NG108-15
Mouse × Rat hybrid neuroblastoma NG108-15 cells were obtained from Sigma-Aldrich. The cells were grown in Dulbecco's Modified Eagle Medium (DMEM, Thermo Fisher Scientific) plus 10% fetal bovine serum (FBS, Gibco), 100 U/ml penicillin, and 100 U/ml streptomycin. The cells were cultured into a humidified incubator (5% $CO_2$, 37°C).

### Hippocampal and DRG neurons
Hippocampal and DRG neurons were obtained from Wistar rats (P2-P3) as described in *Galvanetto, 2018a*. In short, the animals were anesthetized with $CO_2$ and sacrificed by decapitation. The dissociated cells were plated at a concentration of $4×10^4$ cells/ml onto glass round coverslips (170 µm in thickness) coated with 0.5 mg/ml poly-D-lysine (Sigma-Aldrich, St Louis, MO) for 1 hr at 37°C and washed three times in deionized water. It is fundamental to obtain an optimal adhesion of the cells to prevent detachment in the next step (isolation of the cell membrane). The medium used for hippocampal neurons is in Minimum Essential Medium (MEM) with GlutaMAX supplemented with 10% FBS (all from Invitrogen, Life Technologies, Gaithersburg, MD), 0.6% D-glucose, 15 mM HEPES, 0.1 mg/ml apo-transferrin, 30 µg/ml insulin, 0.1 µg/ml D-biotin, 1 µM vitamin B12 (all from Sigma-Aldrich), and 2.5 µg/ml gentamycin (Invitrogen). The medium used for DRG neurons is Neurobasal medium (Gibco, Invitrogen, Milan, Italy) supplemented with 10% FBS (from Invitrogen, Life Technologies, Gaithersburg, MD).

### Rods
Rod cells were obtained from adult male *Xenopus laevis* as described in *Mazzolini et al., 2015*. Under infrared illumination, the eyes of dark-adapted frogs after anesthesia with MS-222 were surgically extracted. Eyes were preserved in the Ringer's solution (110 NaCl, 2.5 KCl, 1 $CaCl_2$, 1.6 $MgCl_2$, 3 HEPES-NaOH, 0.01 EDTA, and 10 glucose in mM; pH 7.8 buffered with NaOH), and hemisected under a dissecting microscope. The extracted retina was maintained in the Ringer's solution.

## Cell transfection

NG108-15 cells were transiently transfected with 300 ng of each cDNA expression plasmids (see *Supplementary file 1*) by using Lipofectamine 2000 Transfection Reagent (Thermo Fisher Scientific) according to its handbook. Briefly, the plasmids (*Supplementary file 1*) and the Lipo2000 were diluted into Opti-MEM Reduced Serum Medium (Gibco), respectively. Five minutes later, we added the diluted DNA to the diluted Lipo2000 to make the plasmid DNA-lipid complexes. After incubating 30 min, we plated the cells on the 12 mm round coverslips coated with ×1 Poly-D-Ornithine (Sigma-Aldrich) in 12-well plate, and in the meanwhile, we added DNA-lipid complexes to the cells. We performed membrane isolation and confocal experiments about 56 hr after transfection.

## Isolation of cell membranes

### Single-cell unroofing (for cell types that grow in adhesion)

The apical membrane of hippocampal neurons, DRGs, and NG108-15 cells were isolated with an optimized version of the unroofing method (*Galvanetto, 2018a*). Briefly, additional empty glass coverslips (24 mm in diameter, 170 µm in thickness) were plasma cleaned for 15 s and broken in four quarters (with the use of the hands) in order to obtain optically sharp edges, as described in *Galvanetto, 2018a*. The coverslip quarters were immersed into 0.5 mg/ml poly-D-lysine for 30 min, and then they were immersed in deionized water for 10 s before use. A Petri dish was filled with Ringer's solution (3.5 ml) without any glucose, where the glass quarter was placed tilted of 7–15 degrees in the middle of it, supported by a 10×10×1 mm glass slice and Vaseline. The cover of the Petri dish was then fixed on the stage of the AFM-inverted microscope setup (JPK Nanowizard 3 on an Olympus IX71).

The cell culture was then mounted on a 3D printed coverslip holder connected to the head stage of the AFM. The AFM head was put on top of the stage in measurement position. The cell culture was immersed into the solution and a target cell was identified and aligned with the underlying corner of the glass quarter. The cell culture was moved toward the corner of the underlying glass with the motors of the AFM until the target cell was squeezed and it doubled its area. At this point the cell is kept squeezed for 3 min, then a loaded spring under the AFM is released to abruptly separate the corner from the cell culture, and break the target cell membrane. (Other applications of scanning probe microscopy with cells or membranes/2D layers can be found here; *Schouteden et al., 2015*; *Xu et al., 2020*.) The glass quarter with the isolated cell membrane was laid down and fixed on the Petri dish. The medium was replaced with Ringer's solution without exposing the cell membrane to the air.

### Membrane isolation of non-adherent cells

Cells that do not grow in adhesion usually do not establish a tight binding with the substrate on top of which they are deposited. For these cells (e.g. rod cells), instead of unroofing, it is more reliable to break the cells with a lateral flux of medium (*Clarke et al., 1975*).

Isolated and intact rods were obtained by mechanical dissociation of the *Xenopus* retina in an absorption buffer (150 mM KCl, 25 mM $MgCl_2$, and 10 mM Trizma base; pH 7.5); they were then deposited on cleaved mica as described in *Maity et al., 2017*. Incubated rods were maintained for 30–45 min over the mica in order to be adsorbed by its negatively charged surface. In the meanwhile, the position of the rods in the field of view of the microscope was annotated. The absorption buffer was substituted by a solution containing (in mM): 150 KCl, 10 Tris-HCl (pH 7.5), and then a lateral flux of medium was applied to the rods until all the cell bodies were removed.

Oocytes overexpressing CNG-N2b were prepared as previously described (*Arcangeletti et al., 2013*) and SMFS membrane fragments were isolated following the protocol of *Maity et al., 2015*.

### Isolation of rod discs

Purification techniques with multiple centrifugations are usually required to isolate membrane-only organelles like rod discs or outer membrane vesicles (*Thoma et al., 2018*). Rod discs were obtained starting from the extracted retina as described in *Maity et al., 2017*. Briefly, discs were separated with two series of centrifugations of the sample overlaid on a 15–40% continuous gradient of OptiPrep (Nycomed, Oslo, Norway). Forty µl of the sample were diluted with 40 µl of absorption buffer, and incubated on freshly cleaved mica for 40 min. After 40 min, the incubation medium was removed and substituted with the solution used in the AFM experiments (150 mM KCl, 10 mM Tris-HCl, pH 7.5).

## Reconstitution of purified membrane proteins

SthK channel was purified as described in *Marchesi et al., 2018*. bR and ChR1 proteins were bought from Cube-Biotech. The purified protein or the different mixtures (as indicated in the Results section) were brought at a total concentration of 0.5 mg/ml in a buffer containing 20 mM HEPES, pH 7.8, 150 mM KCl, and 0.1% *N*-dodecyl β-D maltoside (DDM), and aliquoted into 100 μl samples. A lipid mixture of 1,2-dioleolyl-*sn*-glycero-3-phosphocholine, 1,2-dioleolyl-*sn*-glycero-3-phosphoethanolamine, 1,2-dioleolyl-*sn*-glycero-3-phospho-L-serine at 8:1:1 ratio (Avanti Polar Lipids) was added at a lipid-to-protein ratio of 1 (wt/wt). The different ternary mixtures of protein-lipid-detergent were soni-cated in an ice-bath sonicator for 2 min and subsequently equilibrated on an orbital shaker (200 rmp) for 2 hr. Detergent was removed by hydrophobic adsorption adding twice ~5 mg of wet bio-beads SM-2 and equilibrating under gentle shaking (200 rmp) for 1 hr at RT and overnight at 4°C, respectively. Before use samples were diluted 1:3 in adsorption buffer (20 mM HEPES, pH 7.8, 150 mM KCl).

The reconstituted membrane proteins were adsorbed on freshly cleaved mica for 30 min in a humid chamber. The samples were gently rinsed with imaging buffer (150 mM KCl, 10 mM HEPES, pH = 7.8) and subsequently used for AFM imaging and unfolding.

## Gene silencing experiment

The genes of TRPC1/5/6 and TMEM16F were edited using CRISPR/Cas9 and their sgRNAs were cloned into LentiCRISPR-V2 puro vector (*Supplementary file 1*). 3000 ng of the indicated plasmids were transfected into the NG108-15 cells by using the Lipo2000; 52 hr later after transfection, the cells were used for western blot. In order to select the silencing cell for single-cell unroofing, we treated the cells with culture medium containing 2 μl/ml puromycin (*Jiao et al., 2018*) after 22 hr transfection. After about 68 hr selection of puromycin, the alive cells were subjected to unroofing and the unroofed membrane were used for unfolding experiment.

## AFM imaging and SMFS

AFM experiments was performed using an automated AFM (JPK Nanowizard 3) with 50 μm long cantilevers (AppNano HYDRA2R-NGG, nominal spring constant = 0.84 N/m). We calibrated the AFM cantilevers in the experimental medium before each experiment using the equipartition theorem (*Butt et al., 1995*). The AFM experiments of hippocampal neurons and DRGs were performed with Ringer's solution (NaCl 145 mM, KCl 3 mM, $CaCl_2$ 1.5 mM, $MgCl_2$ 1 mM, HEPES 10 mM, pH 7.4). The AFM experiments of NG108-15 cell membranes were performed in the $Ca^{2+}$-free Ringer's solution supplemented with 10 mM TCEP (Cat# C4706, Sigma). The TCEP can breakdown the disulfide bonds. Rod membrane and discs experiments were performed with 150 mM KCl, 10 mM Tris-HCl, pH 7.5. The fishing of the reconstituted membrane proteins (ChR1 and SthK) and bR was performed with imaging buffer (150 mM KCl, 10 mM HEPES, pH 7.8). All experiments were performed at 24°C.

### AFM imaging

The position of the cells before unroofing was annotated in the monitor of the computer in order to start the AFM imaging where the cells were in contact with the substrate (cell membrane is not visible in bright-field). The membrane obtained with single-cell unroofing (hippocampal neurons, DRG, and NG108-15) can be easily found in proximity to the glass corner (~80% success rate). In the case of the rod membrane (non-adherent cells), usually different positions need to be scanned before finding a patch of membrane. Rod discs and the reconstituted protein patches can be identified only via AFM imaging. We performed imaging both in contact mode (setpoint ~0.4 nN) and in intermittent-contact mode (lowest possible), but the intermittent-contact mode is preferable because it does not damage the border of the patches of membrane.

### AFM-based SMFS (protein unfolding)

We performed automated SMFS on top of the imaged membranes by setting grid positions for the approaching and retraction cycles of the cantilever. All experiments were performed with a retraction speed of 500 nm/s (for hippocampal neurons, DRG, rods and discs), 600 nm/s (for NG108-15 cells), 1000 nm/s (for bR, ChR1, and SthK). The membrane proteins present in the sample were attached non-specifically to the cantilever tip by applying a constant force of ~1 nN for 1 s between the AFM tip and the cytoplasmic side of the membrane. This method proved to work with different membrane

proteins (*Tanuj Sapra et al., 2006*; *Thoma et al., 2017*; *Thoma et al., 2012*), and to allow a higher throughput compared to methods that involve a specific attachment between the tip and the protein (*Cisneros et al., 2005*; *Kedrov et al., 2004*; *Müller et al., 2002*; *Oesterhelt et al., 2000*; *Sumbul et al., 2018a*; *Sumbul et al., 2018b*). Also, in order to demonstrate the sawtooth patterns truly represent the unfolding of membrane proteins, we performed unfolding experiment on the top of the coverslip coating with the poly-D-lysine.

## Western blot

The transfected NG108-15 cells were harvested in the lysis buffer (150 mM NaCl, 1.0% Triton X-100, 0.1% SDS, 0.5% sodium deoxycholate, 10 mM NaF, 1 mM $Na_3VO_4$, 1.0% NP-40, and 40 mM Tris-HCl, pH = 8.0) supplemented with protease inhibitor (Cat# 469311001, Roche) on ice. The samples were boiled at 100°C for 30 min and then the supernatants were loaded in 10% SDS-PAGE gel for gel electrophoresis. The proteins on the gels were transferred onto the PVDF membranes (Cat# 88520, Thermo Fisher Scientific) and these membranes were blocked with 5% nonfat dry milk. The membranes were incubated with the corresponding primary antibodies 1:5000 for α-Tubulin (Cat# T8203, Sigma); 1:500 for TRPC1 (Cat# ACC-101, Alomone Labs)/5 (Cat# ACC-020, Alomone Labs)/6 (Cat#ACC-120, Alomone Labs) and 1:200 for TMEM16F (Cat# ACL-016, Alomone Labs) at 4°C overnight with gentle shaking. Secondary antibodies Goat-anti-rabbit HRP (Cat# P0448, Dako) and Goat-anti-mouse (Cat# P0447, Dako) 1:5000 in 5% BSA were used. Membranes were washed with PBST three times and developed with an ECL-HRP (Cat# WBULS0100, Millipore) system. The gray value ratios of protein bands were quantified using ImageJ software.

## Confocal experiments

### NG108-15 whole cells

The transfected cells and non-transfected cells were fixed by using 4% PFA for 15min at RT. The transfected cells were stained with Hoechst (Life Technologies). The non-transfected cells were permeated by 0.05% Triton X-100 for 4 min and blocked by 10% FBS+5% BSA for 90 min, followed by the incubation of corresponding primary antibodies (1: 250) at 4°C overnight. The samples were rinsed with pre-cold ×1 PBS three times and incubated with the secondary antibody Alexa 594-labeled goat anti-rabbit (1:800, Cat# A11037, Invitrogen) at RT for 90 min. After staining the samples with Hoechst, the samples were tested by Nikon A1R microscope with ×60 oil immersion objective (NA 1.40). The results were analyzed with the ImageJ software.

### NG108-15 cell membranes

The NG108-15 cells were stained with 200 μM SiR-Actin, 200 μM SiR-Tubulin (Spirochrome) or 25 nM MitoTracker Red FM (Thermo Fisher Scientific) for 30 min in a humidified incubator (5% $CO_2$, 37°C). The stained or transfected cells were unroofed according to the method as mentioned above. Then, the NG108-15 cell membranes were tested by Nikon A1R microscope with ×20 objective, followed by the imaging scan by AFM. The results were analyzed with the ImageJ software.

## Molecular visualization

We aligned the current protein structures of TRPC superfamily using the PyMOL software, such as human TRPC3, mouse TRPC4, mouse TRPC5, and human TRPC6. Their codes in PDB are 5zbg, 5z96, 6aei, and 5yx9, respectively. We zoomed the C-terminal part (from S5 to the C-termini) of these structures and find more details.

## Automatic classification of SMFS data

The selection of the F-D curves that represent the unfolding of membrane proteins is usually based on the search for a specific pattern of unfolding in the SMFS data, after a filtering based on the length of the protein under investigation (*Marsico et al., 2007*; *Spoerri et al., 2018*). In the case of a native preparation (like ours) that contains a mixture of unknown proteins: (i) the filtering based on the distance cannot be applied and (ii) the number of specific patterns to be found is unknown. In order to find recurrent patterns of unfolding in an SMFS dataset we developed an algorithm that consists of five major blocks (*Figure 1—figure supplement 9a*). In the first block, the parts of the F-D curves not related to the unfolding process are removed, and a coarse filtering aimed at the detection of

spurious traces is performed. In the second block, a quality score based on the consistency of the experimental data with the WLC model is computed and assigned to each trace. This score is used to select physically meaningful traces for further analysis. In the third block, distances between pairs of traces are computed to assess their similarity. The distances are used in the fourth block for density peak clustering (DPC). The fifth and final block consists in the refinement and possibly in the merging of some of these clusters. In what follows we provide a detailed overview of each block.

## Block 1: Filtering

The standard F-D curve preprocessing was applied to all the data within 'Fodis' (**Galvanetto, 2018b**). The zero of the force of the curve was determined averaging the non-contact part (baseline after the final peak) and subtracted to all the points of the curve. The piezo position was transformed in tip-sample-separation considering the contribution of the bending of the tip to the extension of the polymer. Given that the F-D curves are subject to noise (due to thermal fluctuations, coming from the instrument, etc.), we smooth the original signals through interpolation on a grid with width $\delta_{interp}$ = 1 nm.

A curve is discarded if it does not contain a:

- detectable contact point (i.e. a transition from negative forces to positive forces in respect to the baseline set at zero force);
- if the points occupy force ranges over 5000 pN.

Some of the F-D curves show deviations from the horizontal zero-force line in the non-contact part (wavy final part due to imperfect detachment of the polymer or other noise from the environment). We detect and discard these traces by computing the standard deviation of the tails from the zero-force line. If it exceeds two times $\sigma_{NOISE}$ (average standard deviation of the baseline of the batch of curves), the trace is discarded.

## Block 2: Quality score

The quality score is used to refine selection of traces with high information content vs. noisy traces. It is based on the description provided by the WLC model, which is the standard model in the analysis of SMFS data (**Ainavarapu et al., 2007**). The WLC model implies the equation:

$$F\left(x\right) = \frac{k_B T}{l_p} \left( \frac{1}{4} \left( 1 - \frac{x}{L_c} \right)^{-2} + \frac{x}{L_c} + \frac{1}{4} \right)$$

(1)

where $F$ is force, $x$ is extension, $k_B$ is Boltzmann's constant, $T$ is temperature, $l_p$ is persistence length, and $L_c$ is contour length. Each unfolding curve in the trace is fitted with the WLC equation and an $L_c$ value, corresponding to the length of the unfolded protein domain is obtained. The $L_c$ values are computed by solving **Equation 1** for each $x$ and $F$. An appropriate value for the persistence length $l_p$ for membrane proteins is 0.4 nm as reported in **Ainavarapu et al., 2007**. The WLC model is applicable in the force range 30–500 pN (**Petrosyan, 2016**).

Once we compute the $L_c$ values, we can build an $L_c$ histogram. Normally, the $L_c$ histogram describing a successful unfolding experiment is characterized by the presence of a few maxima separated by deep minima. We implement these features in the definition of our quality score to distinguish meaningful F-D curves.

An important parameter is the bin width of the $L_c$ histogram. If the bin width is too small, the histogram is noisy; if the bin width is too large, peaks corresponding to the unfolding of different domains might be merged. We use bin width 8 nm which is an efficient value for evaluating the goodness of a curve and it allows to consider also curves that deviate from the WLC model ($l_p$ = 0.4 nm) but that contain information. Furthermore, the choice of such large bin width is based on visual inspection of the histograms of proteins with known structure. Once the $L_c$ histogram is built, we detect all maxima and minima. A maximum is meaningful if it is generated by more than five points and it includes more than 1% of the force measures of a trace.

For each maximum in the $L_c$ histogram, we compute a score $W$ quantifying the consistency of the peak with the WLC model. A high-quality peak is clearly separated from other peaks of the histogram, therefore it should be surrounded by two minima. We define $f_{left} = \frac{P_{left}}{P_{max}}, f_{right} = \frac{P_{right}}{P_{max}}$, where $P_{max}$, $P_{left}$, and $P_{right}$ are the probability densities of the maximum, of the left and the right minima. Ideally,

$f \sim \frac{1}{2}(f_{left} + f_{right})$ should go to 0. We define the peak score as $W = exp\left(-2f^2\right)$. According to this definition, if $P_{left} = 1, P_{right} = 2$ and $P_{max}$ =16, W=0.98. While if $P_{left} = 13, P_{right} = 14$, the peak doesn't fit well with the WLC model and W=0.24.

Once a score is computed for each relevant peak in the $L_c$ histogram, that score is assigned to all points in the corresponding trace. This is accomplished in two steps: first, the peak score is assigned to all points in the histogram belonging to that peak. Second, to all points with force values below 30 pN, for which an $L_c$ values cannot be computed due to the model's limitations. To these points, we assign the score of the first successive point with force larger than 30 pN. This criterion applies only to points within 75 nm from the last point assigned to the peak. The peak width value is selected by visual inspection of traces, evaluating the maximum width of their force peaks.

The quality score of a trace, $S_w$, is the sum of the scores for all points in the trace. The higher the global score, the higher the trace quality. We use the ratio between the quality score and the trace length to select high-quality traces. If this ratio is below 0.5, we discard the trace. We assume that if more than half of the trace is inconsistent with the WLC model, it is a low-quality trace and as such we exclude it from the analysis. While if more than half of the trace is in good agreement with the WLC model, it is possibly a meaningful trace.

We point out that the goal of blocks 1–4 is only to find dense recurrent patterns in the SMFS data: in block 5 we reevaluate the F-D curves to form the selections shown in **Figure 3** of the main text.

## Block 3: Computing distances

In block 3 we quantify the similarity between the traces in order to find the recurrent pattern of unfolding within the data. To accomplish this goal, we use a modified version of the distance introduced by **Marsico et al., 2007**. This distance is defined using the dynamic programming alignment score computed for a pair of traces. For two traces, a and b, the distance $d_{ab}$ is simply:

$$d_{ab} = 1 - \frac{S_D\left(N_a, N_b\right)}{N_{max}} \qquad (2)$$

where $S_D\left(N_a, N_b\right)$ is the global alignment score, $N_a$ is the length of trace a, $N_b$ is the length of trace b, and $N_{max}$ is the maximum length between the two. We have modified the match/mismatch scoring function used by Marsico et al. as follows:

$$M(i,j) = \begin{cases} 1 - \frac{|F_a(i) - F_b(j)|}{F_{scoring}} & \text{if } |F_a(i) - F_b(j)| < F_{scoring} \\ \frac{-|F_a(i) - F_b(j)|}{F_{scoring}} & \text{otherwise} \end{cases} \qquad (3)$$

where $F_a(i)$ and $F_b(j)$ are the forces in points i and j in traces a and b, and $F_{scoring} = 4\sigma_{NOISE}$. In the work done by Marsico et al., $F_{scoring}$ is replaced by $\Delta F_{max}$, which is the average of the maximum force values in the two traces. When two widely different traces have high $\Delta F_{max}$ their distance will be lower with respect to two traces with low $\Delta F_{max}$ but overall higher level of similarity. Namely, the distance magnitude depends on the $\Delta F_{max}$ value and traces with high $\Delta F_{max}$ have by definition lower distance values. It is important to note that this problem did not occur in Marsico's work since the $\Delta F_{max}$ values were uniformly distributed for all traces.

In order to gain computational efficiency, the distance is computed only for traces which differ by no more than two peaks in the $L_c$ histograms or by no more than 20% in their trace length difference.

## Block 4: Density peak clustering

The DPC algorithm (**Rodriguez and Laio, 2014**) is used for clustering. This choice is appropriate given that a fraction of traces in the analyzed dataset correspond to statistically isolated events and DPC automatically excludes the outliers. DPC can be summarized in the following steps:

1. We compute the density of data points in the neighborhood of each point using the k-nearest neighbor (k-NN) density estimator (**Altman, 1992**). The density is the ratio between k and the volume occupied by the k-NN:

$$\widetilde{\rho}_i = \frac{k}{\omega_d r_{k,i}^d} \qquad (4)$$

where $d$ is the intrinsic dimension (ID) of the dataset (*Facco et al., 2017*), $\omega_d$ is the volume of the $d$-sphere with unitary radius, and $r_{k,i}$ is the distance of point $i$ from its $k$th nearest neighbor. In DPC it is the density rank which is relevant for the final cluster assignment. Therefore, without loss of generality, we compute the density using the following equation:

$$\rho_i = -log r_{k,i} \tag{5}$$

$\widetilde{\rho_i}$ and $\rho_i$ are related by a simple monotonic transformation and thus, have the same rank. By using *Equation 4* we don't have to compute the ID of the dataset. In order to assign bigger weight to high-quality traces, we multiply $\rho_i$ by the score-length ratio of trace $i$.

2. Next, we find the minimum distance between point $i$ and any other point with higher density, denoted as $\delta_i$ :

$$\delta_i = \min_{j:\ \rho_j > \rho_i} d_{ij} \tag{6}$$

where $d_{ij}$ is the distance between points $i$ and $j$. $\delta_i$ is used to identify the local density maxima.

3. We identify the cluster centers as density peaks, for example, points with high values of both $\rho_i$ and $\delta_i$ . For each point we compute the quantity $\gamma_i = \rho_i \delta_i$ . Points with high values of $\gamma_i$ are good cluster center candidates. We sort all points by the value of $\gamma_i$ in descending order. The first point is a cluster center. The second point is a cluster center unless its distance from the first point is smaller than $r_{cut} = 0.3$ (which represents the distance below which on average two traces are considered as the same pattern). Regarding the third point, it is a cluster center if it is at a distance smaller than $r_{cut}$ from the preceding two points. Following the same logic, all the points are assessed and all cluster centers are identified.

4. All points that are not cluster centers are assigned to the same cluster of the nearest point with higher density.

## Block 5: Refinement and merging

The previous blocks, from 1 to 4, were optimized for finding the centers of dense patterns of unfolding in the SMFS data, but not for finding the borders of the clusters. To solve this issue, that is, finding the F-D curves that are similar to each pattern of unfolding, we used the conventional definition of similarity (degree of superposition of F-D curves in the force/tip-sample-separation plane) automated in the Fodis software in the tool 'fingerprint_roi' (*Galvanetto, 2018b*).

In brief, we superimposed each cluster center with its two closest neighbors creating the effective 'area of similarity' (AoS) for each cluster. The AoS is defined as the area generated by all the points of the three curves above 30 pN and before the last peak (see *Figure 1—figure supplement 9b*), each point forming a square of 5 nm × 5 pN. Then, the SMFS curves are preliminary filtered based on their length with their final peak falling between 0.7 × $L$ and 1.3 × $L$ (with $L$ length of the cluster center). Each of the remaining F-D curves is compared with the AoS, and the number of its points that fall within the AoS is annotated: this number constitutes the similarity score. As depicted in *Figure 1—figure supplement 9c*, the plot of the scores in descending order interestingly forms a line with two different slopes. The change of the slope empirically defines a threshold that reflects the limit of similarity for each cluster. If two clusters share more than 40% of the traces above the threshold, they are considered the same cluster, thus merged (all the merges are reported in *Figure 1—figure supplement 9d*). We merge the clusters in decreasing order: when four consecutive clusters are merged in previous ones and the number of traces in each cluster of the remaining ones is less than 3, we stop the merging and we determine the cutoff number of the clusters in that dataset.

## Formal derivation of the Bayesian identification of the clusters

Bayesian inference is widely used in modern science (*Heenan and Perkins, 2018*; *Wang et al., 2019*) because it allows to univocally determine the level of uncertainty of a hypothesis (*Jaynes, 1986*). We used the same framework to determine the molecular identity of the unfolding clusters. In the most general terms, we observed the unfolding cluster $C_X$ , and we want to find the probability that the unfolding of a certain protein $Prot_A$ corresponds to the unfolding cluster $C_X$ , that is, we want to find the posterior probability $P\left(Prot_A \vee C_X\right)$ . In the form of the Bayes theorem:

$$P\left(Prot_A|C_X\right) = \frac{P\left(C_X|Prot_A\right)P\left(Prot_A\right)}{P\left(C_X\right)} \tag{7}$$

where $P\left(Prot_A\right)$ is the prior, that is, the probability of $Prot_A$ to be in the sample; $P\left(C_XProt_A\right)$ is the likelihood, that is, the likelihood to find a cluster with the features of $C_X$ coming from the unfolding of $Prot_A$ ; and $P\left(C_X\right)$ is the normalizing factor. In the case of a classical experiment with a single purified protein, $P\left(Prot_A|C_X\right)$ is assumed to be equal to 1, but this is not the case for a native environment where there are $Prot_B$ , $Prot_C$ , etc.

The observables of an unfolding cluster for which we determined the likelihood functions are the contour length of the last detectable peak $Lc_{max,Cx}$ (~length of the F-D curve), the peak profile of the cluster in the $L_c$ space $Peaks_{Cx}$ (also called unfolding barriers), and the average unfolding force of the detected peaks $F_{Cx}$ , but the method is modular and therefore it will incorporate also other observables when available. *Equation 7* becomes:

$$P\left(Prot_A|Lc_{max,Cx}, \bar{F}_{Cx}\right) = \frac{P\left(Lc_{max,Cx}|Lc_{Prot_A}\right)P\left(\bar{F}_{Cx}|\bar{F}_{Prot_A}\right)P\left(Peaks_{Cx}|Structure_{Prot_A}\right)P\left(Prot_A\right)}{N} \tag{8}$$

where $N = \sum_i(P\left(Lc_{max,Cx}|Lc_{Prot_i}\right)P\left(F_{Cx}|F_{Prot_i}\right)P\left(Peaks_{Cx}|Structure_{Prot_A}\right)P\left(Prot_i\right))$ is the normalizing factor that takes into consideration all the proteins $Prot_i$ present in the sample. In the next paragraphs we will describe the determination of the numerator of *Equation 8*.

### Determination of prior $P\left(Prot_A\right)$

The most crucial part of the method is the determination of the list of proteins present in the sample, together with all their properties (length, abundance, secondary structure, topology, etc.). To do so we combined the mass spectrometry results of the cells under investigation (*Chen et al., 2006*; *Kwok et al., 2008*; *Panfoli et al., 2008*) with other structural and topological information available in Uniprot and PDB. The crossing of the databases is done, thanks to the unique Uniprot identifier. The complete list of proteins of hippocampal neurons, rod outer segments, and rod discs with the information necessary for the Bayesian inference are shown in *Supplementary file 3* of the article. In case the data of the species of interest are not available, cross species proteomic analysis demonstrated that the majority of proteins are conserved in terms of relative abundance (*Bayram et al., 2016*; *Wright et al., 2010*).

$P\left(Prot_A\right)$ is the probability of finding $Prot_A$ and not $Prot_B$ , $Prot_C$ , etc., which corresponds to the normalized relative abundance of $Prot_A$ in the sample – a parameter that is usually calculated in mass spectrometry analysis. Indeed, in silico calculation of abundances gives rather trustworthy values:

1. The most accurate option is the emPAI (*Ishihama et al., 2005*).
2. If the emPAI is not available, the second best option is the spectra counting for each peptide (PSM) (*Liu et al., 2004*).
3. If the PSM is not available, the sequence coverage can be used as loose estimation (*Florens et al., 2002*).

We used the emPAI for hippocampal neurons and rods; for the discs, the emPAI does not give accurate values because of the extreme concentration of Rhodopsin, therefore we used the abundances obtained with other quantitative methods (*Milo and Phillips, 2016*).

We demonstrated in *Figure 1—figure supplements 2–4* that the isolated patches of membrane contain the membrane proteins of the original cells but not the cytoplasmic proteins, therefore we created an additional binary variable *ismembrane* for each protein, equal to 0 if the protein is not a membrane protein, 1 otherwise. This information is extracted from the annotation in the Uniprot database. The final prior is:

$$P\left(Prot_A\right) = abundance_A \times ismembrane_A$$

### Modeling the conditional probability $P\left(Lc_{max,Cx}Lc_{Prot_A}\right)$

The F-D curves encode a reliable structural information, that is the total length of the unfolded protein (*Oesterhelt et al., 2000*). We revisited 14 published unfolding clusters of membrane proteins (*Bosshart et al., 2012*; *Cisneros et al., 2005*; *Ge et al., 2016*; *Kawamura et al., 2013*; *Kedrov et al., 2004*; *Klyszejko et al., 2008*; *Maity et al., 2015*; *Möller et al., 2003*; *Oesterhelt et al., 2000*;

*Sapra et al., 2009*; *Serdiuk et al., 2016*; *Thoma et al., 2017*; *Thoma et al., 2012*) plus our own experiments, and that allowed us to create the conditional probability function for the observable $Lc_{max,Cx}$ as shown in *Figure 4B–C*. The distribution of the effective unfolding lengths forms a Gaussian bell centered at $0.89Lc_{Prot_A}$ with a standard deviation of $\sigma_{effective} = 0.05Lc_{Prot_A}$ . The final standard deviation is obtained combining $\sigma_{effective}$ with the error due to the unperfect determination of the persistence length $\sigma_p = 0.07Lc_{Prot_A}$ , and the error of the $Lc_{max,Cx}$, that is, $\sigma_{Lc} = 0.05Lc_{Prot_A}$ , so we obtained $\sigma_{Likelihood} = \sqrt{\sigma_{effective}^2 + \sigma_p^2 + \sigma_{Lc}^2} = 0.10Lc_{Prot_A}$ . This prior is determined without considering any assumption on the pinning point of the protein (i.e. the position of the amino acid fixed at the level of the membrane that determine $Lc_{max,Cx}$), just considering the experimental data.

Most of these experimental data comes from membrane proteins that do not have a final cytoplasmic/extracellular domain. These data show also that the $0.89Lc_{Prot_A}$ correction of *Figure 4c* originates from a pinning point that can occur anywhere between the last amino acids of the last transmembrane domain and the penultimate loop (*Bosshart et al., 2012*; *Möller et al., 2003*; *Thoma et al., 2017*). There are only few examples of SMFS of proteins with a large cytoplasmic domain at one terminal (*Maity et al., 2015*; *Tanuj Sapra et al., 2006*), and these data suggest that the pinning point to the membrane or to another protein can be (i) at the terminus end of the protein, (ii) everywhere along the domain, or (iii) at the end of the transmembrane segments. In other words, the pinning point of a protein that has a final domain cannot be firmly established if not within the domain range. Therefore, for the proteins that have topological annotations, we replaced the Gaussian prior described above with a non-informative prior normalized to 1 and constant between $(L_{protein} - L_{finaldomain}) < Lc_{max,Cx} < L_{protein}$, 0 elsewhere.

We use this non-informative prior only for proteins with $L_{finaldomain} \geq 2\sigma_{Likelihood}$ ; for proteins with small final domains $L_{finaldomain} < 2\sigma_{Likelihood}$ , we used the Gaussian prior centered at $0.89Lc_{Prot_A}$ . We chose the threshold of $2\sigma_{Likelihood}$ in order to take into account the uncertainty on the position of the pinning points and the errors $\sigma_p$ and $\sigma_{Lc}$ that are still present even in proteins with small final domains.

## Determination of the conditional probability $P\left(Peaks_{Cx}Structure_{Prot_A}\right)$

Empirical evidence (*Bosshart et al., 2012*; *Cisneros et al., 2005*; *Ge et al., 2016*; *Kawamura et al., 2013*; *Kedrov et al., 2004*; *Klyszejko et al., 2008*; *Maity et al., 2015*; *Möller et al., 2003*; *Oesterhelt et al., 2000*; *Sapra et al., 2009*; *Serdiuk et al., 2016*; *Thoma et al., 2017*; *Thoma et al., 2012*) and simulations (*Yamada et al., 2016*) suggest that the occurrence of the peaks is more likely in presence of unstructured regions, that is, in the intracellular or extracellular loops of the membrane proteins (see *Figure 4f*). We used this information to evaluate the probability that the structure of the $Prot_A$ unfolds with the unfolding pattern of cluster $C_X$ (position of the peaks). To do so, we calculated the cross-correlation between $Peaks_{Cx}$, that is, the global histogram of cluster $C_X$ (of point between 40 and 100 pN for normalization purposes) and $Structure_{Prot_A}$, that is, the profile of Gaussian bells centered in the center of the loops of $Prot_A$ and with an FWHM of 15 nm which is an average estimate of the distance between two loops (see *Figure 4—figure supplement 1*). If no structure is available we cross-correlated a flat line. Both $Structure_{Prot_A}$ and $Peaks_{Cx}$ are normalized to have a total area under the profile equal to 1. Then, we assigned to $P\left(Peaks_{Cx}Structure_{Prot_A}\right)$ the maximal value of the cross-correlation with a relative lag of ±15 nm to allow some freedom for fine alignment.

## Determination of the conditional probability $P\left(F_{Cx} \vee F_{Prot_A}\right)$

The force necessary to unfold a protein domain depends on the stability of the domain itself. α-Helices and β-sheets are unfolded at different force levels as shown in *Figure 4d*. We revised the unfolding forces of 22 proteins (*Bosshart et al., 2012*; *Cisneros et al., 2005*; *Ge et al., 2016*; *Hoffmann and Dougan, 2012*; *Kawamura et al., 2013*; *Kedrov et al., 2004*; *Klyszejko et al., 2008*; *Maity et al., 2015*; *Möller et al., 2003*; *Oesterhelt et al., 2000*; *Sapra et al., 2009*; *Serdiuk et al., 2016*; *Thoma et al., 2017*; *Thoma et al., 2012*) and we used as $P\left(F_{Cx} \vee F_{Prot_A}\right)$ the smoothed trend line of the distribution (*Figure 4e* of the main text).

## Assessment of the total number of collected traces needed to generate a cluster

We provide here a first-order estimate on the relation between the abundance of a protein and the number of pulling traces needed to generate a reliable cluster of F-D curves of that protein (50 similar traces are usually sufficient to generate a reliable cluster).

We consider the case of the well-characterized rod disc sample. A rod disc has an area of ~3 µm$^2$ and the number of Rhodopsins in a disc is ~$10^5$ (**Arnadóttir and Chalfie, 2010**) so the protein concentration is ~$3*10^4$ Rhodopsin/µm$^2$. We can use as an example the most abundant cluster in the rod disc (the DISC1, $N_{1c}$ = 285) where the total number of traces collected was $N_{tot}$ = 106,528. Assuming linearity, we can write

$$N_{1c} = K N_{tot} \left[P\right]_{2D}$$

where $N_{1c}$ is the number of traces in one cluster, $N_{tot}$ is number of collected traces, $\left[P\right]_{2D}$ is the protein concentration, and $K$ is the pulling efficiency factor (with units of µm$^2$). By solving this equation for Rhodopsin in the rod discs we find $K = 9*10^{-8}$µm$^2$. In first approximation we can then conclude that in order to generate a cluster of $N_{1c}$ = 50 traces, the product $N_{tot} \left[P\right]_{2D}$ (number of collected traces * protein concentration) needs to be greater than $5.6*10^8$ traces*protein/µm$^2$.

## Acknowledgements

We thank Alexander Bruce for useful comments on the manuscript. We thank Dr Kosaku Shinoda for support in the emPAI calculation. We thank Prof. Anna Menini who provided the mTMEM16A-GFP and mTMEM16F-GFP plasmids. We thank Prof. Guidalberto Manfioletti who provided the peGFP-N1 plasmid.

## Additional information

### Funding
No external funding was received for this work.

### Author contributions
Nicola Galvanetto, Conceptualization, Data curation, Software, Writing – original draft; Zhongjie Ye, Arin Marchesi, Simone Mortal, Sourav Maity, Data curation; Alessandro Laio, Software; Vincent Torre, Conceptualization, Project administration

### Author ORCIDs
Nicola Galvanetto  http://orcid.org/0000-0002-0408-1747
Zhongjie Ye  http://orcid.org/0000-0003-0306-5267
Arin Marchesi  http://orcid.org/0000-0002-8219-1642
Simone Mortal  http://orcid.org/0000-0001-6534-9324
Vincent Torre  http://orcid.org/0000-0001-8133-3584

### Decision letter and Author response
Decision letter https://doi.org/10.7554/eLife.77427.sa1
Author response https://doi.org/10.7554/eLife.77427.sa2

## Additional files

### Supplementary files
- Supplementary file 1. Tables with list of plasmids and sgRNA sequences.
- Supplementary file 2. Sample statistics (experiments and clusters).
- Supplementary file 3. Tables with proteins find with mass spectrometry.
- Transparent reporting form

## Data availability

Data not present as supplementary data are available in https://github.com/galvanetto/NativeSMFS/releases (copy archived at swh:1:rev:96e51a937818e80436dc53d73c7f33f949de7963), as well as the open source software to read them.

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
