## [Editor Report]

This paper presents a method to identify membrane proteins in native cell membranes based on a combination of single molecule AFM and an unsupervised clustering procedure to identify clusters of single-protein curves. This original approach represents a definitive step forward for AFM technology and methodology, which can generally only be used to characterize purified biomolecules of known identity.

---

## [Decision Letter]

**Decision letter after peer review:**

Thank you for submitting your article "Unfolding and identification of membrane proteins in situ" for consideration by *eLife*. Your article has been reviewed by 3 peer reviewers, and the evaluation has been overseen by a Reviewing Editor and Richard Aldrich as the Senior Editor. The following individuals involved in review of your submission have agreed to reveal their identity: Aaron T Blanchard (Reviewer #2); Rafael Tapia-Rojo (Reviewer #3).

The 3 reviewers and I have considered that your approach represents a very interesting method to identify proteins in a native membrane environment. We suggest some points that should be improved to reinforce your manuscript, but we do not request new experiments.

Essential revisions:

1. The utility of the technique should be better motivated. Its advantages, limitations and future directions (in particular, considering the low yield) should be discussed more thoroughly.

2. Although we think that the computational analysis does effectively take care of some issues listed below, could you explicitly address the following points in your discussion:

a) Can you confirm that shortened traces are discarded at the computational stage, eliminating those corresponding to attachment at arbitrary positions or loss of tip-protein interaction during the experiment? How do you take care of discrepancies between the contour length and the length measured by mass spectroscopy?

b) What happens if the protein has different conformations corresponding to different force traces?

c) What happens if the protein interacts with other proteins, which could influence the traces?

3. Some concerns were raised about the reproducibility of the technics. The number of samples, number of cantilevers, number of curves for each sample, number of clustered curves found by the computational algorithm, etc… should all be reported explicitly in the supplementary information.

*Reviewer #1 (Recommendations for the authors):*

Statistical reporting should be improved. For each clustered dataset that is presented, the number of total curves, filtered curves, curves of various classes etc… should all be reported.

The software/code for the clustering and analysis algorithms should be made publicly available, along with the raw datasets.

Figure S2: the precise method for cell 'unroofing' should be schematically diagrammed better so as to allow others to attempt to reproduce it. The exact procedure is not clear to me.

Figure S4 shows force (y axis) vs. contour length (x-axis). What is the equation they use for this transformation? It should be reported or a reference provided

Figure S11: How is the cutoff on the number of clusters decided? i.e., for DRG why were the top 15 clusters taken and not the top 10 or top 18?

*Reviewer #2 (Recommendations for the authors):*

To be more specific about my general critiques: The introduction does not paint a particularly clear vision of the specific types of new knowledge that could be gained from this technique that could not be obtained from existing approaches. As such, I did not feel particularly excited until I reached the Results section. Furthermore, by the end of the Results section, I didn't feel convinced that the unique capabilities of the technique had been demonstrated. The subsection at the end of the Results section, titled "structural insights from SMFS", provides one interesting insight using the large dataset. However, in my opinion, the authors don't effectively link this insight to their new technique. How did this new technique lead to this structural insight in a way that wouldn't have been possible otherwise? Furthermore, for a study that unveils a new method such as this, I would expect one or two more subsections that illustrate additional pieces of knowledge that could be obtained thanks to this new approach. I don't think that the insight presented is interesting or useful enough to stand alone as the only new piece of scientific knowledge presented in this paper.

*Reviewer #3 (Recommendations for the authors):*

I believe that the manuscript deserves publication. I have some comments, mostly related to recognizing the method's limitations and envisioning potential directions to overcome (or at least mitigate) them. The manuscript would benefit from a richer discussion in this direction.

1. The authors constantly speak about protein unfolding or rupture peaks, but I believe that this is not correct or at least precise. Different from in vitro experiments pulling on proteins tethered to the substrate, here, the mechanical resistance arises from pulling helical groups out of their native membrane environment. This should be specified. Further, there is very little (or almost no) comment on how the magnitude of the force peaks correlate with the expectations (if there can be any expectation). For example, in Fig. 6, the expected "rupture regions" are guessed based on the contour length increments arising from pulling out that protein fragment, right? Do the observed forces relate to the expected forces, given that some proteins exhibit a considerable heterogeneity of peak heights (for example, TRPC6). Can the authors elaborate on the mechanical dimension of their identification method? Can this information be combined with the contour length increments to deliver an additional parameter for protein identification?

2. The authors mention that a limitation of the pipeline is the possibility of merging in the same cluster of different proteins that could exhibit similar force-extension patterns. I think that, in addition to this, another limitation could arise from proteins that exhibit heterogeneous patterns, meaning that they can be pulled out of the membrane in different ways. Have the authors accounted for this or observed protein clusters that could correspond to the same protein species? I guess that this situation could be identified by inspecting clusters with the same total contour length and maybe recurrent patterns (perhaps this heterogeneity can arise only when pulling out a fraction of the protein, likely those involving lower forces). Please comment on this.

3. As the authors acknowledge, a limitation of the method is the very low yield, as they can only identify a tiny fraction of the total membrane proteome. Can the authors comment on potential ways to improve this? Developing some chemical strategies for achieving specific pickups could help in this direction. Non-specific AFM is highly ineffective, and there are many developments to overcome this limitation in single-molecule experiments. For example, proteins can be covalently anchored to substrates from their N-terminus with the appropriate chemistry. By implementing a similar strategy with their cantilever, the fraction of membrane proteins pulled from their N-terminus could be dramatically enriched. I'm not asking that the authors do this, but at least comment on how the experimental procedure could be improved to achieve a higher experimental yield.

4. Could this pipeline be extended to identify cytoplasmatic proteins? Can the authors discuss this possible future direction?

---

## [Author Response]

Essential revisions:1. The utility of the technique should be better motivated. Its advantages, limitations and future directions (in particular, considering the low yield) should be discussed more thoroughly. (See Reviewers #2 and 3).

We added new paragraphs in the Introduction, Results and dedicated sections in the Discussion where the overall motivation and limitations are discussed more accurately. In particular we better described the possibility of obtaining the membrane proteomic profile from a very limited amount of biological material (sub-femtogram samples) compared to what is currently possible with Mass spectrometry that requires at least thousands of cells. Reviewer #2 also provided an important suggestion on an aspect that we didn’t stress enough in the previous version of the manuscript, i.e. that with this work we increased more than 2-fold the number of SMFS spectra available from the literature, and that native SMFS could be an alternative to conventional SMFS where limiting factors like protein purification hinder the production of SMFS data on membrane proteins.

2. Although we think that the computational analysis does effectively take care of some issues listed below, could you explicitly address the following points in your discussion:a) Can you confirm that shortened traces are discarded at the computational stage, eliminating those corresponding to attachment at arbitrary positions or loss of tip-protein interaction during the experiment?

Shortened traces (lack of at least one force peak) originating from attachment or detachment at arbitrary position are not clustered together with the full unfolding of the protein. If the shortened traces present clean peaks and the raising part properly follow the Worm like chain model, they could pass the quality filter (see Block 2 of the methods). But since they are arbitrary their distance between other traces will be large and therefore they don’t form a populated cluster.

We answered to this in Line 394 (and following Lines) of the main text.

How do you take care of discrepancies between the contour length and the length measured by mass spectroscopy?

We answered to this in Line 398 (and following Lines).

b) What happens if the protein has different conformations corresponding to different force traces?

We answered to this in Line 367, 415 (and following Lines).

c) What happens if the protein interacts with other proteins, which could influence the traces?

The traces are almost certainly influenced by other proteins, but this is actually what we want. Imagine for instance a tetrameric ion channel like the CNG (Maity et al., 2015): it is only thanks to the combined structure that the CNG has a pore, and an isolated monomer is not functional. So in order to get some relevant information for the structure in the tetramer, you need to unfold the protein in the tetramer. In this way the trace will capture the stabilizing effect of the surrounding monomers together with the lipids. The same applies for a ‘G-protein coupled receptor’ type of complex.

We answered to this in Line 375 (and following Lines).

3. Some concerns were raised about the reproducibility of the technics (See Reviewer #1). The number of samples, number of cantilevers, number of curves for each sample, number of clustered curves found by the computational algorithm, etc… should all be reported explicitly in the supplementary information.

We added this information in Supplementary file 2 and referred to it in the main text.

Reviewer #1 (Recommendations for the authors):Statistical reporting should be improved. For each clustered dataset that is presented, the number of total curves, filtered curves, curves of various classes etc… should all be reported.

We added this information in Supplementary File 2 and referred to it in the main text.

The software/code for the clustering and analysis algorithms should be made publicly available, along with the raw datasets.

We included the algorithms, the clustering code and the dataset together with the notes for using them in https://github.com/galvanetto/NativeSMFS/releases/tag/v1.0 and we referred to it in the Methods.

Figure S2: the precise method for cell 'unroofing' should be schematically diagrammed better so as to allow others to attempt to reproduce it. The exact procedure is not clear to me.

We included a new supplementary figure with more details in the caption (Figure 1—figure supplement 1).

Figure S4 shows force (y axis) vs. contour length (x-axis). What is the equation they use for this transformation? It should be reported or a reference provided

We added the details in the caption of Figure 1—figure supplement 5.

Figure S11: How is the cutoff on the number of clusters decided? i.e., for DRG why were the top 15 clusters taken and not the top 10 or top 18?

All the traces that pass the Quality filter (see Block 2 in Methods) are clustered with the density peak clustering procedure. The number of the resulting clusters at this point depends on the actual distance matrix and cannot be predicted a priori (see Block 3 and 4 in Methods). Usually, the number of clusters is in the order of 20-30. The clusters are ordered depending on the number of traces they contain: the first cluster has the largest number of traces, the last clusters are usually populated by 1 or 2 traces. Then there is the final step of clustering (see Block 5). This last step is very important because the density peak clustering is optimized to find the cluster centers, but it is not very accurate determining the edge of the clusters (traces that are assigned within the cluster but with low score). So we used the similarity in *tip-sample separation* VS *force* space to find the end of the clusters (Figure 1—figure supplement 9 b) to refine the population within one cluster. We also introduced a merging procedure: if two clusters share more than 40% of the traces above the threshold determined with the fingerprint_roi procedure (Figure 1—figure supplement 9 b and c), they are considered the same cluster, thus merged (all the merges are reported in Figure 1—figure supplement 9 d). The cutoff (i.e. the final number of clusters for each sample) is determined as follows: we check the clusters in decreasing order, when 4 consecutive clusters are merged and the number of traces in each cluster of the remaining ones is less than 3, we stop the merging and we determine the cutoff. Various threshold parameters have been tested in these processes. The rationale for choosing these values was that the final results remained mostly constant around the chosen thresholds. We revised this description in Block 5 of the Methods.

Reviewer #2 (Recommendations for the authors):To be more specific about my general critiques: The introduction does not paint a particularly clear vision of the specific types of new knowledge that could be gained from this technique that could not be obtained from existing approaches. As such, I did not feel particularly excited until I reached the Results section. Furthermore, by the end of the Results section, I didn't feel convinced that the unique capabilities of the technique had been demonstrated. The subsection at the end of the Results section, titled "structural insights from SMFS", provides one interesting insight using the large dataset. However, in my opinion, the authors don't effectively link this insight to their new technique. How did this new technique lead to this structural insight in a way that wouldn't have been possible otherwise?

We thank the Reviewer#2 for these stimulating criticisms. We have now expanded the motivations for the approach we are proposing in the main text.

An example: the mechanical stability of TRPC6 in its native membrane where it is most likely surrounded by a G-protein coupled receptor and other partners is different (and closer to native) than the mechanical stability of a purified TRPC6 and reconstituted in an artificial membrane. Therefore we believe that the structural insights that can be gathered with our approach can be an important complement to other approaches that require expression and purifications, and we are not aware of other techniques that can do the same.

Furthermore, for a study that unveils a new method such as this, I would expect one or two more subsections that illustrate additional pieces of knowledge that could be obtained thanks to this new approach. I don't think that the insight presented is interesting or useful enough to stand alone as the only new piece of scientific knowledge presented in this paper.

In the new version of the manuscript, we improved the description of the utility and the results that can be achieved.

Reviewer #3 (Recommendations for the authors):I believe that the manuscript deserves publication. I have some comments, mostly related to recognizing the method's limitations and envisioning potential directions to overcome (or at least mitigate) them. The manuscript would benefit from a richer discussion in this direction.1. The authors constantly speak about protein unfolding or rupture peaks, but I believe that this is not correct or at least precise. Different from in vitro experiments pulling on proteins tethered to the substrate, here, the mechanical resistance arises from pulling helical groups out of their native membrane environment. This should be specified.

We are not completely sure about the statement “[in membrane proteins] the mechanical resistance arises from pulling helical groups out of their native membrane”. For instance Yu et al. (Yu et al., 2017) showed that little segments of alpha helices (so secondary structures) can unfold and fold back under tension within the membrane (see their Fig. 1d, insert; or the perspective (Müller and Gaub, 2017)). We specified this problem in Line 131 (and following Lines).

Further, there is very little (or almost no) comment on how the magnitude of the force peaks correlate with the expectations (if there can be any expectation). For example, in Fig. 6, the expected "rupture regions" are guessed based on the contour length increments arising from pulling out that protein fragment, right? Do the observed forces relate to the expected forces, given that some proteins exhibit a considerable heterogeneity of peak heights (for example, TRPC6). Can the authors elaborate on the mechanical dimension of their identification method? Can this information be combined with the contour length increments to deliver an additional parameter for protein identification?

Predicting a priori the unfolding force of a segment is to our knowledge not possible but a statistical prediction is possible and indeed in our Bayesian method we included the conditional probability extracted from data in literature. Beta sheets tend to unfold with higher rupture forces than alpha helices (see Figure 4d-e). We use this information to improve our predictions.

2. The authors mention that a limitation of the pipeline is the possibility of merging in the same cluster of different proteins that could exhibit similar force-extension patterns. I think that, in addition to this, another limitation could arise from proteins that exhibit heterogeneous patterns, meaning that they can be pulled out of the membrane in different ways. Have the authors accounted for this or observed protein clusters that could correspond to the same protein species? I guess that this situation could be identified by inspecting clusters with the same total contour length and maybe recurrent patterns (perhaps this heterogeneity can arise only when pulling out a fraction of the protein, likely those involving lower forces). Please comment on this.

We answered to this in Line 367, 415 (and following Lines).

3. As the authors acknowledge, a limitation of the method is the very low yield, as they can only identify a tiny fraction of the total membrane proteome. Can the authors comment on potential ways to improve this? Developing some chemical strategies for achieving specific pickups could help in this direction. Non-specific AFM is highly ineffective, and there are many developments to overcome this limitation in single-molecule experiments. For example, proteins can be covalently anchored to substrates from their N-terminus with the appropriate chemistry. By implementing a similar strategy with their cantilever, the fraction of membrane proteins pulled from their N-terminus could be dramatically enriched. I'm not asking that the authors do this, but at least comment on how the experimental procedure could be improved to achieve a higher experimental yield.

Thanks, we commented this issue in Lines 427 and following.

4. Could this pipeline be extended to identify cytoplasmatic proteins? Can the authors discuss this possible future direction?

In order to perform SMFS we need an anchoring point for the protein. The cell membrane is a non perfect but very effective anchoring for membrane proteins; native cytoplasmic proteins lack an anchoring point. Moreover, it is evident from our control experiments (See Figure 1—figure supplement 2,3,4) that cytoplasmic proteins don’t remain in the surface. We can think of an extension of our technique that could allow to expose the extracellular side of the cell membrane, but not to perform SMFS on cytoplasmic proteins (unless properly mutated and purified). We describe this idea in the last paragraph of the Discussion (line 436).

We thank the reviewers for their very helpful requests.

References

Arnadóttir J, Chalfie M. 2010. Eukaryotic mechanosensitive channels. Annu Rev Biophys 39:111–137. doi:10.1146/annurev.biophys.37.032807.125836

Kawamura S, Gerstung M, Colozo AT, Helenius J, Maeda A, Beerenwinkel N, Park PS-H, Müller DJ. 2013. Kinetic, Energetic, and Mechanical Differences between Dark-State Rhodopsin and Opsin. Structure 21:426–437. doi:10.1016/j.str.2013.01.011

Kessler M, Gaub HE. 2006. Unfolding Barriers in Bacteriorhodopsin Probed from the Cytoplasmic and the Extracellular Side by AFM. Structure 14:521–527. doi:10.1016/j.str.2005.11.023

Maity S, Mazzolini M, Arcangeletti M, Valbuena A, Fabris P, Lazzarino M, Torre V. 2015. Conformational rearrangements in the transmembrane domain of CNGA1 channels revealed by single-molecule force spectroscopy. Nature Communications 6:7093. doi:10.1038/ncomms8093

Müller DJ, Engel A. 2007. Atomic force microscopy and spectroscopy of native membrane proteins. Nat Protoc 2:2191–2197. doi:10.1038/nprot.2007.309

Müller DJ, Gaub HE. 2017. Membrane proteins scrambling through a folding landscape. Science 355:907–908. doi:10.1126/science.aam8370

Oesterhelt F, Oesterhelt D, Pfeiffer M, Engel A, Gaub HE, Müller DJ. 2000. Unfolding Pathways of Individual Bacteriorhodopsins. Science 288:143–146. doi:10.1126/science.288.5463.143

Yu H, Siewny MGW, Edwards DT, Sanders AW, Perkins TT. 2017. Hidden dynamics in the unfolding of individual bacteriorhodopsin proteins. Science 355:945–950. doi:10.1126/science.aah7124